# SENSITIVITY-AWARE VISUAL PARAMETER-EFFICIENT TUNING

## ABSTRACT

Visual Parameter-efficient Tuning (VPT) has become a powerful alternative for full fine-tuning, which only updates a small number of parameters while freezing the remaining vast majority of parameters to significantly reduce the storage costs for adapting the pre-trained vision models to downstream tasks. Although the storage burden is largely alleviated, VPT approaches still face many challenges, e.g., lower inference speed and lacking effective configurations for trainable parameters tailored for each task. In this paper, we present a simple yet effective approach termed **S**ensitivity-aware visual **P**arameter-efficient **T**uning (SPT) to tackle these challenges. Given a desired tunable parameter budget, SPT quickly identifies the important parameters to the given task in a data-dependent way before fine-tuning, without the complex selection schedule. Then, SPT adaptively determines the tuning granularity for each weight matrix. Accordingly, for the whole model, we structurally tune the entire sensitive weight matrices that contain a large proportion of sensitive parameters (structured tuning), and non-structurally tune the sensitive connections in the insensitive weight matrices (unstructured tuning), simultaneously. For structured tuning, SPT approximates the update with the low-rank reparameterization to preserve the parameter budget. Therefore, our SPT has high flexibility and representational capability while achieving favorable trade-off between parameter-efficiency and accuracy. Through extensive experiments on a wide range of downstream recognition tasks, our SPT achieves better overall transfer performance than the full fine-tuning and the other VPT approaches, with no additional computational or memory overhead during inference. For instance, SPT saves 99.35% of the trainable parameters than the full fine-tuning while achieving a 7.3% higher average top-1 accuracy on VTAB-1k benchmark with the supervised pre-trained ViT-B backbone. Notably, SPT is also the first work that bridges the gap between full fine-tuning and VPT approaches with backbones under self-supervised pre-training strategies MAE and MoCo v3 on the challenging VTAB-1k benchmark.

## 1 INTRODUCTION

The pre-training and fine-tuning paradigm has underpinned the most recent breakthroughs in vision, yielding stunning empirical performance on a series of tasks such as segmentation (Chen et al., 2017; Ronneberger et al., 2015) and detection (He et al., 2017; Carion et al., 2020). Transformer (Vaswani et al., 2017) has been widely adopted as the standard architecture for pre-trained vision models, with representatives including CLIP (Radford et al., 2021), MAE (He et al., 2022b), BEiT (Bao et al., 2022), etc. To effectively adapt the pre-trained representations to the downstream tasks, the de-facto choice is fine-tuning, which initializes the model with the pre-trained weights and tunes all the parameters. However, vanilla fine-tuning needs to store a separate instance of parameters for each task and each deployment scenario. It can be extremely storage-intensive as the storage cost grows linearly with the number of possible cases, considering there are vast varieties of downstream tasks and dynamic deployment environments, especially when deploying the large vision models (Dosovitskiy et al., 2021; Liu et al., 2021; Xu et al., 2021b) to mobile systems. For example, even storing a single large pre-trained ViT-H (He et al., 2022b) model on a local disk requires at least 2.3GB, while the top-10 U.S. apps required only collectively 2.2GB in May 2021.[1]

---

[1] https://sensortower.com/blog/ios-app-size-growth-2021

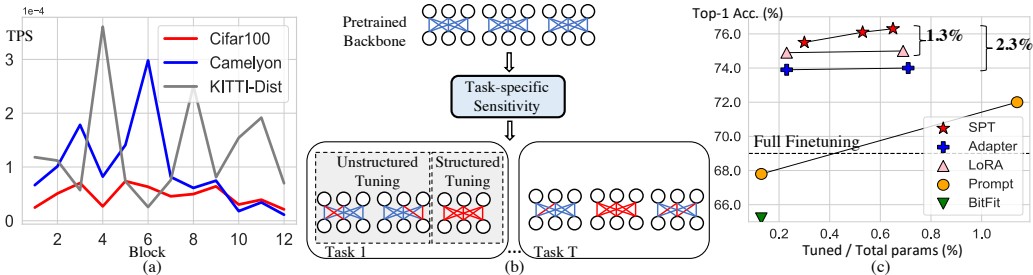

Figure 1: (a) The block-wise parameter sensitivity with supervised pre-trained ViT-B backbone (Dosovitskiy et al., 2021) for three sampled tasks from VTAB-1k (Zhai et al., 2019a). "TPS" denotes our task-specific parameter sensitivity (importance). We show averaged scores over all 800 training samples. The sensitivity of each block varies markedly across different tasks. (b) Our proposed Sensitivity-aware visual Parameter-efficient Tuning (SPT) identifies the task-specific important positions and adaptively combines unstructured and structured tuning to enjoy both flexibility and high capacity. The blue and red lines represent the frozen and trainable parameters, respectively. (c) Accuracy vs. parameter efficiency with the supervised pre-trained ViT-B backbone. Our SPT has no extra computational overhead during inference, surpasses full fine-tuning by large margins, and performs favorably against other VPT approaches.

Notably, an emerging trend is to replace the full fine-tuning with Visual Parameter-efficient Tuning (VPT) (Jia et al., 2022; Chen et al., 2022; Zhang et al., 2022), which only tunes a small number of trainable parameters (newly introduced or inherently in the model) to cooperate with a frozen backbone that is shared by multiple tasks. As VPT approaches exhibit less than 1% of the trainable parameters, the storage burden is largely alleviated. Another attractive property of VPT is that tuning fewer parameters eases the optimization difficulty and mitigates the overfitting issue for the large models, thereby achieving comparable or even better performance than fine-tuning (Jia et al., 2022) (see Figure 1 (c)). Although promising, the existing VPT approaches suffer from two major issues. First, they specify the positions to add the trainable parameters with different heuristics, and the importance of these positions has not been well studied. For instance, Prompt tuning (Jia et al., 2022) and Adapter (Houlsby et al., 2019) add trainable parameters to the input space and each Transformer (Vaswani et al., 2017) block, respectively. Moreover, these approaches keep the same configuration for the trainable parameters across different downstream tasks, neglecting their domain gaps and characteristics. Second, the additional parameters lead to a non-negligible sacrifice on the inference efficiency in terms of speed and memory consumption. Taking Prompt tuning (Jia et al., 2022) as an example, with the enlarged input space (200 prompts), it exhibits 2× slower inference speed and consumes 2× of the GPU memory than the full fine-tuning counterpart.

To this end, in this work, we present a novel Sensitivity-aware visual Parameter-efficient Tuning (SPT) that identifies and tunes the parameters at *task-specific important positions* while being *inference-efficient*. Based on the assumption that not all pre-trained parameters contribute equally to the performance across different tasks, we first propose a new criterion to efficiently measure the sensitivity (importance) of the pre-trained backbone parameters to a specific task for our SPT. Inspired by model pruning methods (Srivastava et al., 2015; Molchanov et al., 2019), we propose to use loss reduction for the sensitivity measurement, which can be efficiently approximated with a first-order Taylor expansion. The resulting parameter sensitivity is solely computed from the gradients, and therefore it can be quickly derived ahead of fine-tuning. We show an example of parameter sensitivities using a pre-trained ViT-B (Dosovitskiy et al., 2021) backbone in Figure 1 (a), where the sensitivities vary across different tasks.

Next, an intuitive solution is to only tune the parameters with the highest sensitivity, which we name as unstructured tuning following (Han et al., 2015; 2016). Despite its simplicity and flexibility, unstructured tuning still lacks representational capability as only a few parameters are tuned to capture the domain gap. To this end, our SPT further incorporates unstructured tuning with structured tuning (Figure 1 (b)). Specifically, after identifying the sensitive parameters of the pre-trained backbone, SPT adaptively determines the tuning granularity for each weight matrix. Accordingly, for the whole model, we structurally tune the entire sensitive weight matrices that contain a large proportion of sensitive parameters (structured tuning), and non-structurally tune the sensitive connections in the insensitive weight matrices (unstructured tuning), simultaneously. To preserve the parameter budget, for structured tuning, SPT follows the efficient reparameterization strategy of LoRA (Hu

et al., 2022) to optimize the low-rank decomposition. Therefore, our SPT merits high flexibility and representational capability from both tuning granularities. After fine-tuning, our SPT only needs to store a subset of task-specific weights and indexes, which can be merged into the backbone, thereby being parameter-efficient while having no extra cost during inference.

This paper has the following key contributions. 1) We introduce a sensitivity criterion to measure the importance of the pre-trained backbone parameters, which is fast, effective, and can be applied to backbones with various pre-training strategies. 2) Based on the sensitivity criterion, we propose a parameter-efficient tuning approach to tune parameters at task-specific important positions, which includes not only unstructured tuning but also structured tuning to achieve high flexibility, large capacity, and favorable tradeoff between parameter-efficiency and accuracy. 3) Extensive experiments on a total of 24 downstream recognition tasks with vision Transformer backbones under supervised, MAE (He et al., 2022b), and MoCo v3 (Chen et al., 2021) pre-trainings show that our SPT achieves the overall best performance, outperforming vanilla fine-tuning and the other SOTA VPT methods by clear margins (Figure 1 (c)). Moreover, to the best of our knowledge, SPT for the first time bridges the gap between full fine-tuning and VPT approaches on VTAB-1k under MAE and MoCo v3 self-supervised pre-trainings.

## 2 RELATED WORK

The full fine-tuning is the most predominant approach when adapting a large-scale pre-trained model to downstream tasks, where the model is initialized from the pre-trained weights with all parameters trainable. Yet, when a model becomes larger, parameter-efficient tuning (Lester et al., 2021; Li & Liang, 2021) is highly desirable, which transfers a pre-trained model to the downstream tasks by tuning only a tiny portion of parameters to alleviate the storage burden. The general parameter-efficient tuning approaches can be categorized into addition-based methods (Jia et al., 2022; Houlsby et al., 2019; He et al., 2022a; Chen et al., 2022) and reparameterization-based methods (Zaken et al., 2022; Hu et al., 2022; Xu et al., 2021a; Guo et al., 2021; Xu et al., 2021a).

*Addition-based methods* attach additional trainable parameters to the backbone and only tune these parameters. Apart from Prompt tuning (Jia et al., 2022) and Adapter (Houlsby et al., 2019), recent addition-based methods study connecting or combining existing VPT methods. For instance, He et al. (2022a) connect Prompt tuning and Adapter and provide a unified view that all VPT approaches share the same design to modify the backbone outputs. Zhang et al. (2022) search for the optimal configurations to combine multiple VPT approaches following once-for-all approaches (Cai et al., 2020; Wu et al., 2021). However, the additional parameters require extra computations compared to the full fine-tuning, thereby introducing additional lags during inference.

*Reparameterization-based methods* tune the parameters that are inherently in the backbone or new parameters that can be merged into the backbone, thereby yielding no extra computational costs during inference. For reparameterization-based methods, one line of work tunes the same set of trainable parameters for different tasks, e.g., tuning the bias terms (Zaken et al., 2022) or the last several layers (Yosinski et al., 2014; Caelles et al., 2017). The representative work LoRA (Hu et al., 2022) optimizes two low-rank matrices to approximate the update of each weight matrix in the self-attention modules, which can be merged into the pre-trained weights during inference. Another line of work explores task-specific trainable parameters by jointly optimizing the model parameters and tuning configurations (Guo et al., 2021; Zhao et al., 2020). For example, Guo et al. (2021) seek task-specific trainable parameters by optimizing a sparse mask with $L_0$ norm and Zhao et al. (2020) optimize binary masks to identify trainable parameters. However, learning the extra masks to identify the tunable parameters requires a formidable amount of training time and GPU memory, which makes these methods hard to be applied to large vision models (Liu et al., 2021; Xu et al., 2021b). Our method also belongs to the reparameterization-based methods but differs from the existing ones in two aspects. On one hand, we follow importance estimation approaches (Molchanov et al., 2019; Lee et al., 2019) and adopt an effective sensitivity criterion to quickly identify the task-specific trainable parameters before fine-tuning. On the other hand, we incorporate both structured and unstructured tuning granularities to enable higher flexibility and representational power than the sole unstructured counterpart, offering consistent accuracy gains for a fixed parameter budget.

Apart from the fine-tuning strategies, the choice of the pre-trained models affects the adaptation performance greatly. Recently, models pre-trained with self-supervised strategies (Caron et al., 2021; He et al., 2020; 2022b; Xiao et al., 2021) have gained much popularity as they better tackle the over-

fitting issue than supervised pre-trained models (He et al., 2020). Although several works (He et al., 2022b; Ericsson et al., 2021) argue better transferability for the self-supervised pre-trained models, their effectiveness is barely studied under the VPT settings. Recently, Jia et al. (2022) show that unlike the supervised pre-trained models, applying the SOTA VPT approaches with representative self-supervised pre-trained strategies, i.e., MAE (He et al., 2022b) and MoCo v3 (Chen et al., 2021), achieves inferior results than the full fine-tuning. AdaptFormer (Chen et al., 2022) achieves higher performance than the vanilla fine-tuning on the video domain with MAE pre-training, while it still falls behind the full fine-tuning in the image domain. In contrast to the previous work, to the best of our knowledge, our SPT for the first time bridges the gaps on the popular VTAB-1k benchmark under the backbones pre-trained with both MAE and MoCo v3.

## 3    METHOD

To introduce our Sensitivity-aware visual Parameter-efficient Tuning (SPT), we first describe our simple yet effective criterion to measure the task-specific sensitivity for pre-trained backbone parameters in Section 3.1. We then introduce how SPT employs the task-specific sensitive parameters to adaptively incorporate both unstructured and structured tuning granularities, while having no additional computational overhead during inference in Section 3.2.

### 3.1    TASK-SPECIFIC PARAMETER SENSITIVITY

Based on our assumption that not all parameters contribute equally to the performance across different tasks, we first propose a new criterion to measure the sensitivity for the parameters in the pre-trained backbone, that is specifically tailored for a given task.

Specifically, given the training dataset $\mathcal{D}_t$ for the $t$-th task and the pre-trained model weights $\boldsymbol{w} = \{w_1, w_2, \ldots, w_N\} \in \mathbb{R}^N$ where $N$ is the total number of parameters, the objective for the task is to minimize the empirical risk: $\min_{\boldsymbol{w}} E(\mathcal{D}_t, \boldsymbol{w})$. We denote the sensitivity criterion as $\mathcal{S} = \{S_1, \ldots, S_N\}$. The sensitivity $S_n$ for parameter $w_n$ is measured by the empirical risk difference when tuning it

$$S_n = E(\mathcal{D}_t, \boldsymbol{w}) - E(\mathcal{D}_t, \boldsymbol{w} \mid w_n = w_n^*), \tag{1}$$

where $w_n^* = \underset{w_n}{\mathrm{argmin}}(E(\mathcal{D}_t, \boldsymbol{w}))$. We can reparameterize the fine-tuned parameters as $w_n^* = w_n + \Delta_{w_n}$, where $\Delta_{w_n}$ denotes the change for $w_n$ after tuning. However, it is computationally intensive to compute Eq. (1) for two reasons. Firstly, getting the empirical risk for $N$ parameters requires forwarding the entire network $N$ times, which is too time-consuming. Secondly, it is challenging to derive $\Delta_{w_n}$, as we have to tune each individual $w_n$ until convergence.

To overcome the first barrier, we simplify the empirical loss by approximating $S_n$ in the vicinity of $\boldsymbol{w}$ by its first-order Taylor expansion:

$$S_n^{(1)} = -g_n \Delta_{w_n}, \tag{2}$$

where the gradient $\boldsymbol{g} = \partial E / \partial \boldsymbol{w}$ and $g_n$ is the gradient for the $n$-th element.

To address the second barrier, following (Liu et al., 2019; Cai et al., 2019), we take the one-step unrolled weight as the surrogate for $w_n^*$ and approximate $\Delta_{\boldsymbol{w}_n}$ in Eq. (2) with a single step of gradient descent. We can accordingly get $S_n^{(1)} \approx g_n^2 \epsilon$, where $\epsilon$ is the learning rate. Since $\epsilon$ is the same for all parameters, we can eliminate it and finally get $S_n^{(1)} \approx |g_n|$. This essentially states that the sensitivity for a parameter is solely measured based on the magnitude of its gradient. This is intuitive since the pre-trained weights with higher sensitivity are more likely to affect downstream tasks fine-tuned with SGD. Getting $S$ is also very simple, which can be derived once prior to fine-tuning.

It's noteworthy that it would make only slight differences if we use a higher order Taylor expansion in Eq. (2). Without loss of generality, we take second-order Taylor expansion as an example and get $S_n^{(2)} \approx g_n^2 \epsilon - \frac{1}{2} \epsilon^2 g_n \boldsymbol{H}_n \boldsymbol{g}$, where $\boldsymbol{H}_n$ is the $n$-th row of the Hessian matrix $\boldsymbol{H} = \partial^2 E / \partial \boldsymbol{w}^2 \in \mathbb{R}^{N \times N}$. Since $\epsilon$ is small, the first term is much larger than the second one. Thus, we directly use the first-order Taylor expansion in this paper.

In practice, we accumulate $\mathcal{S}$ from a total number of $C$ training samples to generate accurate sensitivity, where $C$ is a pre-defined hyper-parameter. We derive $\mathcal{S}$ ahead of fine-tuning to simplify the process as shown in Algorithm 1.

---

**Algorithm 1** Computing task-specific parameter sensitivities.

---

**Input:** Pre-trained model with network parameters $w$, training set $\mathcal{D}_t$ for the $t$-th task, and number of training samples $C$ used to calculate the parameter sensitivities.
**Output:** Sensitivity set $\mathcal{S} = \{S_1, \ldots, S_N\}$.
Initialize $\mathcal{S} = \{0\}^N$.
**for** $i \in \{1, \ldots, C\}$ **do**
    Get the $i$-th training sample of $\mathcal{D}_t$.
    Compute loss $E$.
    Compute gradients $g$.
    **for** $n \in \{1, \ldots, N\}$ **do**
        Update the $n$-th sensitivity: $S_n = S_n + |g_n|$.
    **end for**
**end for**

---

### 3.2 SENSITIVITY-AWARE VISUAL PARAMETER-EFFICIENT TUNING

After obtaining the sensitivity set $\mathcal{S}$ and a desired parameter budget $\tau$, a straightforward solution is unstructured tuning, which directly tunes the top-$\tau$ most sensitive unstructured connections (parameters). Specifically, we get the top-$\tau$ sensitive connections to form $\mathcal{T}$ from $\mathcal{S}$. For any weight matrix $W \in \mathbb{R}^{d_{\text{in}} \times d_{\text{out}}}$, we aim to get a binary mask $M \in \mathbb{R}^{d_{\text{in}} \times d_{\text{out}}}$ computed by

$$M^j = \begin{cases} 1 & W^j \in \mathcal{T} \\ 0 & W^j \notin \mathcal{T} \end{cases}, \tag{3}$$

where $W^j$ and $M^j$ are the $j$-th element in $W$ and $M$, respectively. Accordingly, the updated weight matrix can be formulated as $W' \leftarrow W - \epsilon g_W \odot M$, where $g_W$ is the gradient for $W$. In this way, only the sensitive parameters are updated while the other parameters are frozen.

However, considering VPT approaches generally limit the proportion of trainable parameters to less than 1%, tuning only a small number of unstructured connections might not have enough representational capacity to handle the datasets with large domain gaps from the pretraining data. Therefore,

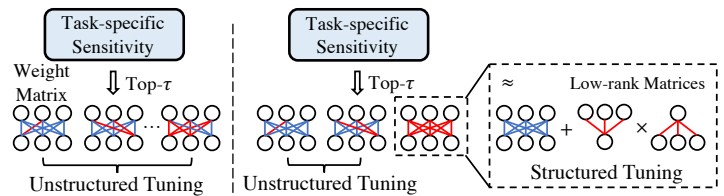

Figure 2: Comparison between unstructured tuning and adaptively combining unstructured and structured tuning granularities. The latter structurally tunes the sensitive weight matrices that a large proportion of their parameters are sensitive. The blue and red lines represent the frozen and trainable parameters.

we propose to improve the representational capability under the same parameter budget by incorporating unstructured tuning with structured tuning and **adaptively** determine the tuning granularity for each weight matrix. In structured tuning, we update the entire sensitive weight matrices instead of only some unstructured connections as depicted in Figure 2. Formally, we employ a pre-defined threshold hyper-parameter $\sigma$ and the updated weight matrix $W'$ with our SPT can be formulated as:

$$W' = \begin{cases} W'_s & \text{if } \frac{\sum_{j=0}^{d_{\text{in}} \times d_{\text{out}}} M^j}{d_{\text{in}} \times d_{\text{out}}} \geq \sigma \\ W - \epsilon g_W \odot M & \text{otherwise} \end{cases}, \tag{4}$$

where $W'_s$ denotes the structurally updated $W$. Eq. (4) essentially states that we structurally tune $W$ when its proportion of the sensitive parameters is higher than $\sigma$. To implement $W'_s$, one can directly tune all the parameters in $W$ and get $W'_s \leftarrow W - \epsilon g_W$. However, directly tuning the entire weight matrix is parameter-consuming, e.g., a weight matrix in the feed-forward layer takes up 2.7% of the total parameters for ViT-B (Dosovitskiy et al., 2021). To preserve the parameter budget when tuning the entire sensitive weight matrices, we freeze the pre-trained weights and reparameterize the update with two trainable low-rank weight matrices in a parameter-efficient way as in LoRA (Hu et al., 2022). Therefore, we implement $W'_s \leftarrow W + W_{\text{down}}W_{\text{up}}$, where $W_{\text{down}} \in \mathbb{R}^{d_{\text{in}} \times r}$ and $W_{\text{up}} \in \mathbb{R}^{r \times d_{\text{out}}}$ are randomly initialized learnable low-rank matrices to approximate the update of $W$ and the rank $r$ is a hyper-parameter that $r \ll \min(d_{\text{in}}, d_{\text{out}})$. By letting $\frac{\sigma}{r} > \frac{d_{\text{in}} + d_{\text{out}}}{d_{\text{in}} \times d_{\text{out}}}$, we ensure that the total number of trainable parameters does not exceed $\tau$ to preserve the parameter

budget. In this way, our SPT incorporates both structured and unstructured tuning granularities adaptively with our sensitivity criterion to enable higher flexibility and stronger representational power simultaneously, leading to noticeable gains in parameter-efficiency vs. accuracy tradeoff. We investigate the effectiveness of both tuning methods in Section 4.3.

**Discussions.** Although our SPT employs LoRA (Hu et al., 2022) when conducting structured tuning, it is fundamentally different from LoRA. First, similar to the other VPT approaches (Jia et al., 2022; Houlsby et al., 2019), LoRA neglects the domain gaps across different downstream tasks and adds the low-rank reparameterizations to positions with heuristics. To tackle this fundamental challenge, our SPT estimates the parameter sensitivity and accordingly identifies the task-specific sensitive positions. We show that applying the VPT modules (i.e., visual prompts (Jia et al., 2022), LoRA (Hu et al., 2022), and adapter modules (Houlsby et al., 2019)) to the task-specific instead of heuristic positions achieves solid performance gains (Section 4.3). Second, LoRA employs only structured tuning that updates entire weight matrices. In contrast, under arbitrary parameter constraints, SPT integrates both structured and unstructured tuning granularities, and adaptively determines the tuning granularity for each weight matrix according to their proportion of sensitive parameters. In this way, our SPT can achieve higher flexibility and stronger representational power simultaneously, leading to noticeable gains in parameter-efficiency vs. accuracy trade-off (Section 4.2) and higher scalability than LoRA (Section D).

## 4 EXPERIMENTS

### 4.1 EXPERIMENTAL SETUP

**Datasets and metrics.** We evaluate our SPT on total 24 downstream tasks in two groups following (Jia et al., 2022). 1) FGVC is a benchmark for fine-grained visual classification, which includes CUB-200-2011 (Wah et al., 2011), NABirds (Van Horn et al., 2015), Oxford Flowers (Nilsback & Zisserman, 2008), Stanford Cars (Gebru et al., 2017), and Stanford Dogs (Khosla et al., 2011) datasets. Each of the FGVC datasets contains between 55 to 200 classes and a few thousands of images in total for train, validation, and test. We follow (Jia et al., 2022) for their validation splits if the validation set is unavailable. 2) VTAB-1k (Zhai et al., 2019b) is a large-scale transfer learning benchmark consisting of a collection of 19 visual classification tasks. VTAB-1k can further be divided into three groups, including Natural tasks with natural images, Specialized tasks with images captured by specialized equipments, e.g., medical images, and Structured tasks with images mostly generated from synthetic environments. Each of the VTAB-1k dataset has only 1,000 training samples, while the test set sizes vary. We use top-1 accuracy (%) averaged within each group as our main metric following (Jia et al., 2022).

**Pre-trained backbones.** We conduct experiments on the representative plain vision Transformer backbone ViT-B/16 (Dosovitskiy et al., 2021) that is pre-trained on ImageNet (Krizhevsky et al., 2012) with different pre-training strategies following (Jia et al., 2022), including supervised pre-training, self-supervised pre-training with MAE (He et al., 2022b) and MoCo v3 (Chen et al., 2021).

**Contenders.** We categorize the baseline methods into two groups which are addition-based and reparameterization-based methods, respectively. Unless specified, all baseline methods keep the backbone frozen. Addition-based methods require extra computations during inference, which includes ADAPTER-$k$ (Houlsby et al., 2019), PROMPT-SHALLOW (Jia et al., 2022) that adds trainable prompts only to the input space, PROMPT-DEEP (Jia et al., 2022) that adds the prompts to each layer, NOAH (Zhang et al., 2022), and MLP-$k$ that replaces the classification head with a trainable $k$-layer multi-layer perceptrons as described in (Jia et al., 2022). Reparameterization-based methods have no additional computational overhead during inference, which includes LINEAR that only tunes the classification head (details in (Jia et al., 2022)), PARTIAL-$k$ that fine-tunes the last $k$ layers (details in (Jia et al., 2022)), BITFIT (Zaken et al., 2022), and LORA-$k$ (Hu et al., 2022). We refer the readers to Section 2 for details of the baseline methods. We also define FULL as the full fine-tuning.

### 4.2 MAIN RESULTS

We first evaluate the effectiveness of our method by comparing SPT with baseline methods. The results with ViT-B under supervised pre-training are presented in Table 1, and those under self-supervised pre-training are presented in Table 2.

Table 1: Comparison with the other visual parameter-efficient tuning methods on FGVC datasets and VTAB-1k datasets using supervised ViT-B/16 pre-trained on ImageNet-21k (Dosovitskiy et al., 2021). "Tuned/Total" denotes the fraction of the trainable parameters. Top-1 accuracy (%) and inference speed (ms/img) are reported.

| ViT-B/16 (85.8M) | Total params | FGVC | | VTAB-1k | | | | | Inference speed |
|---|---|---|---|---|---|---|---|---|---|
| | | Tuned / Total | Mean Acc. | Tuned / Total | Natural | Specialized | Structured | Mean Acc. | |
| FULL | 24.02× | 100% | 88.5 | 100% | 75.9 | 83.4 | 47.6 | 69.0 | 2.8 |
| **Addition-based methods** | | | | | | | | | |
| MLP-3 | 1.35× | 1.50% | 79.8 | 1.42% | 67.8 | 72.8 | 30.6 | 57.1 | 2.9 |
| PROMPT-SHALLOW | 1.04× | 0.31% | 84.6 | 0.13% | 76.8 | 79.7 | 47.0 | 67.8 | 3.7 |
| PROMPT-DEEP | 1.18× | 0.98% | **89.1** | 1.14% | 78.5 | 82.4 | 55.0 | 72.0 | 3.8 |
| ADAPTER-8 | 1.06× | 0.39% | 85.5 | 0.23% | 79.0 | 84.1 | 58.5 | 73.9 | 2.9 |
| ADAPTER-32 | 1.19× | 0.95% | 85.6 | 0.71% | 79.6 | 84.0 | 58.3 | 74.0 | 2.9 |
| NOAH | - | - | - | 0.50% | **80.2** | **84.9** | **61.3** | **75.5** | 3.3 |
| **Reparameterization-based methods** | | | | | | | | | |
| LINEAR | 1.02× | 0.12% | 79.3 | 0.04% | 68.9 | 77.2 | 26.8 | 57.6 | |
| PARTIAL-1 | 3.00× | 8.38% | 82.6 | 8.30% | 69.4 | 78.5 | 34.2 | 60.7 | |
| BITFIT | 1.05× | 0.13% | 88.4 | 0.13% | 73.3 | 78.3 | 44.1 | 65.2 | |
| LORA-8 | 1.07× | 0.55% | 86.0 | 0.23% | 79.5 | 84.6 | 60.5 | 74.9 | 2.8 |
| LORA-16 | 1.18× | 0.90% | 84.8 | 0.69% | 79.8 | 84.9 | 60.2 | 75.0 | |
| SPT (Ours) | 1.13× | 0.66% | 88.7 | 0.53% | **81.8** | 85.8 | 60.8 | 76.1 | |
| SPT (Ours) | 1.17× | 0.91% | **89.2** | 0.65% | **81.8** | **85.9** | 61.1 | **76.3** | |

Table 2: Comparison with the other visual parameter-efficient tuning methods on VTAB-1k datasets using self-supervised ViT-B/16 pre-trained by MAE (He et al., 2022b) and MoCo v3 (Chen et al., 2021). "Tuned / Total" denotes the fraction of the trainable parameters. Top-1 accuracy (%) is reported.

| ViT-B/16 (85.8M) | Total Params | VTAB-1k MAE | | | | VTAB-1k MoCo v3 | | | |
|---|---|---|---|---|---|---|---|---|---|
| | | Tuned / Total | Natural | Specialized | Structured | Mean Acc. | Tuned / Total | Natural | Specialized | Structured | Mean Acc. |
| FULL | 38.02× | 100% | 59.3 | 79.7 | 53.8 | 64.3 | 100% | 72.0 | 84.7 | 42.0 | 69.6 |
| **Addition-based methods** | | | | | | | | | | | |
| ADAPTER-8 | 1.08× | 0.23% | **57.2** | 78.4 | 54.7 | **63.4** | 0.23% | 27.6 | 70.9 | 48.4 | 49.0 |
| ADAPTER-32 | 1.28× | 0.95% | 55.3 | **78.8** | 53.3 | 62.5 | 0.95% | 29.2 | 73.4 | **49.8** | 50.8 |
| PROMPT-SHALLOW | 1.02× | 0.12% | 40.0 | 69.7 | **67.5** | 59.1 | 0.12% | 67.3 | 82.3 | 37.6 | 62.4 |
| PROMPT-DEEP | 1.05× | 0.48% | 36.0 | 60.6 | 26.6 | 41.1 | 0.48% | **70.3** | **83.0** | 42.4 | **65.2** |
| **Reparameterization-based methods** | | | | | | | | | | | |
| LINEAR | 1.02× | 0.12% | 18.9 | 52.7 | 23.7 | 32.1 | 0.12% | 67.5 | 81.1 | 30.3 | 59.6 |
| PARTIAL-1 | 4.16× | 8.30% | 58.4 | 78.3 | 47.6 | 61.5 | 8.30% | 72.3 | **84.6** | 47.9 | 68.3 |
| BITFIT | 1.06× | 0.13% | 54.6 | 75.7 | 47.7 | 59.3 | 0.13% | **72.9** | 81.1 | 53.4 | 69.2 |
| LORA-8 | 1.08× | 0.23% | 57.5 | 77.7 | 57.7 | 64.3 | 0.23% | 21.2 | 66.7 | 45.1 | 44.3 |
| LORA-16 | 1.28× | 0.69% | 57.3 | 77.1 | **59.9** | 64.8 | 0.69% | 16.0 | 64.0 | 48.7 | 42.9 |
| SPT (Ours) | 1.15× | 0.40% | 61.6 | **78.7** | 56.7 | 65.7 | 0.39% | 71.5 | 83.0 | 56.6 | 70.4 |
| SPT (Ours) | 1.20× | 0.56% | **61.7** | **78.7** | 57.8 | **66.1** | 0.51% | 71.7 | 84.1 | **58.6** | **71.5** |

First, SPT *outperforms full fine-tuning by remarkable margins with supervised pre-trained ViT-B*. As shown in Table 1, SPT has 7.3% higher mean top-1 accuracy on VTAB-1k than the full fine-tuning with only 0.65% of the trainable parameters. With abundant data, SPT also outperforms the full fine-tuning by 0.7% mean top-1 accuracy in FGVC datasets while saving 99.09% trainable parameters. Our conjecture is that the full fine-tuning is likely to overfit in our studied scenarios of adapting a large pre-trained model to a small/medium target dataset. Second, SPT achieves *better overall performance than the SOTA VPT methods with supervised pre-trained ViT-B*. For instance, in Table 1, SPT with 0.53% proportion of the trainable parameters outperforms the best-performing addition-based method NOAH with a similar amount of tuned parameters by 0.6% mean top-1 accuracy on VTAB-1k, without affecting the inference efficiency. SPT also achieves a 1.3% top-1 accuracy gain with 0.04% less trainable parameters than LORA-16 that has the highest performance within the reparameterization-based baselines. We speculate that the superiority of our SPT comes from: 1) allocating the trainable parameters to more important task-specific positions with our sensitivity criterion; 2) incorporating both unstructured and structured tuning, which achieves higher flexibility and representational capability. Third, SPT *bridges the gap for self-supervised pre-trained backbones* between the existing VPT approaches and the full fine-tuning as shown in Table 2. VPT approaches generally exhibit inferior results than full fine-tuning with the self-supervised pre-

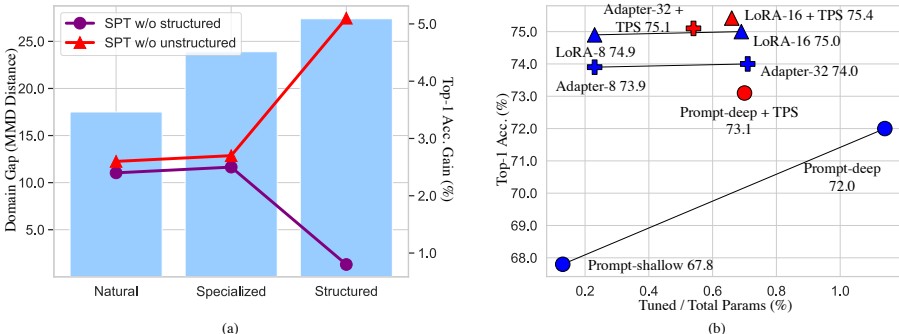

(a)

(b)

Figure 3: (a) The domain gap vs. performance for the three dataset groups in VTAB-1k. We measure the performance by the mean top-1 accuracy gain of our SPT against PROMPT-DEEP (Jia et al., 2022). The domain gap is measured by Maximum Mean Discrepancy (MMD) distance (Tzeng et al., 2014) between the averaged last-layer features of the pre-training images in ImageNet (Krizhevsky et al., 2012) and the fine-tuning images. Employing the structured tuning improves the averaged performance on the Structured datasets significantly. (b) Results for applying other popular VPT modules to the positions identified by our sensitivity criterion on VTAB-1k. "TPS" refers to our task-specific parameter sensitivity. Our criterion brings consistent performance gain.

trained backbones MAE and MoCo v3. We observe that our SPT consistently outperforms the full fine-tuning with both pre-training strategies. Especially, SPT outperforms full fine-tuning for non-neglectable margins of 1.8% and 1.9% top-1 accuracy on VTAB-1k dataset using only 0.56% and 0.51% trainable parameters for MAE and MoCo v3 pre-trained backbones, respectively. Together with our observation in Sections A and B in appendix that: 1) parameters of a pre-trained backbone are with diverse sensitivities to different tasks; 2) self-supervised pre-trained backbones have higher sensitivity variance among the datasets, we conjecture that tailoring the task-specific trainable parameters with SPT can better handle the large variance of the important parameters across tasks and enable accurate adaptation for the self-supervised pre-trained backbones. We report more multi-seed results of our SPT in Table C.

### 4.3 ABLATION STUDY

**Effect of structured and unstructured tuning.** We investigate the effectiveness of combining unstructured tuning with structured tuning described in Section 3.2. The results are presented in Table 3. We observe that under similar fractions of trainable parameters, SPT without unstructured tuning has 0.9% top-1 mean accuracy drop and SPT without structured tuning has a significant 2.4% top-1 mean accuracy drop, suggesting both factors are essential for SPT's superior performance while structured tuning contributes more. Further investigating this phenomenon, we find that the main difference lies in the performance for the Structured datasets in VTAB-1k as shown in Figure 3 (a). Figure 3 (a) also shows that Structured datasets have a larger domain gap to the pre-training dataset than the other datasets (see examples in Section C in the appendix). Our structured tuning updates all channels of a weight matrix efficiently with the low-rank reparameterization, which mitigates the large domain gap for the Structured datasets due to its enhanced representational power.

**Effect of the sensitivity criterion.** We investigate the effectiveness of our sensitivity criterion described in Section 3.1 by employing visual prompts in (Jia et al., 2022), adapter modules in (Houlsby et al., 2019), and LoRA modules in (Hu et al., 2022) to the positions identified by our criterion. We present the results in Figure 3 (b). It can be seen that our criterion brings 1.1%, 1.1%, and 0.4% performance gains for PROMPT-DEEP, ADAPTER-32, and LORA-16, respectively. The consistent performance gains demonstrate the superiority of our sensitivity to select task-specific important positions. It is also indicated that our criterion has wider applications and can be seamlessly combined with the other VPT approaches.

**Effect of the number of training samples $C$ for sensitivity scores.** We investigate the effect of the number of training images $C$ (Algorithm 1). We randomly sample the training samples when $C/N < 1.0$ and report the median over three runs. The results are shown in Table 4. When varying

Table 3: Ablate on the structured tuning and unstructured tuning for SPT on VTAB-1k (Zhai et al., 2019b). Top-1 accuracy (%) is reported. We set different parameter constraints to align the fractions of the trainable parameters for these cases.

| Method | Tuned / Total | Natural | Specialized | Structured | Mean Acc. |
|---|---|---|---|---|---|
| SPT | 0.65% | **81.8** | **85.9** | **61.1** | **76.3** |
| SPT w/o unstructured | 0.66% | 81.1 | 85.1 | 60.1 | 75.4 |
| SPT w/o structured | 0.74% | 80.9 | 84.9 | 55.8 | 73.9 |

Table 5: Cost comparison with full fine-tuning, PROMPT-DEEP (Jia et al., 2022), and LoRA (Hu et al., 2022). PROMPT-DEEP and SPT are evaluated with around 0.70% fractions of the trainable parameters. We report the latency (ms/img) and the peak memory usage (GB).

| Method | Inference Latency (ms/img) | Inference Memory (GB) | Fine-tuning Memory (GB) |
|---|---|---|---|
| FULL | **2.8** | **1.3** | 11.9 |
| PROMPT-DEEP | 3.8 | 1.9 | 13.2 |
| LoRA | **2.8** | **1.3** | 8.2 |
| SPT w/o unstructured | **2.8** | **1.3** | 8.3 |
| SPT | **2.8** | **1.3** | 12.5 |

the ratio from 0.3 to 1.0, we get similar proportions of trainable parameters and find that half of the training samples suffice getting the highest performance.

**Computational cost analysis.** We investigate the computational cost of SPT by comparing it with the full fine-tuning, the addition-based method PROMPT-DEEP (Jia et al., 2022), and LoRA (Hu et al., 2022), which is evaluated on a single GeForce 3090 GPU. The results are presented in Table 5. We observe that PROMPT-DEEP has inevitably higher inference latency and inference GPU memory due to the additional prompts. In contrast, since the updated parameters after fine-tuning can be reparame-

Table 4: Ablate on the number of training samples used to get the sensitivity on VTAB-1k. Top-1 accuracy (%) is reported.

| $C/N$ | 0.3 | 0.5 | 0.7 | 1.0 |
|---|---|---|---|---|
| Mean Acc. | 76.1 | **76.3** | **76.3** | **76.3** |

terized and merged into the pre-trained model, our SPT and LoRA are more efficient than PROMPT-DEEP during inference. We observe that our SPT has higher fine-tuning memory than the full fine-tuning and LoRA which is taken up by masking the gradients in Eq. (4). Nevertheless, compared to the pre-training, which takes thousands of TPUv3-core-days (Dosovitskiy et al., 2021), the fine-tuning process for the total 19 datasets of the VTAB-1k benchmark only takes several GPU hours and is computationally friendly. Therefore, we argue that slightly more fine-tuning memory (0.6 G higher than the full fine-tuning) is affordable and saving fine-tuning memory is not the main purpose of parameter-efficient tuning. Moreover, one can employ LoRA to the task-specific positions found by our SPT (SPT w/o unstructured tuning) to achieve similar fine-tuning memory as LoRA but higher performance (Figure 3 (b)).

## 5 CONCLUSION AND FUTURE WORK

In this paper, we have proposed a novel visual parameter-efficient tuning approach for the adaptation of large pre-trained vision models to downstream tasks. Specifically, we have proposed a criterion to quickly measure the importance of the pre-trained parameters for specific tasks during fine-tuning. Based on the criterion, we have also proposed to tune the task-specific important weight connections for accurate adaptation. To remedy the lack of representational power while achieving favorable tradeoff between parameter efficiency and accuracy, we have proposed to integrate both structured and unstructured tuning, enabling the model to have high flexibility and transferability while being inference-efficient. Experimental results have demonstrated the effectiveness of our proposed SPT on a total of 24 downstream tasks. Notably, we have shown that our approach consistently bridges the gap between vanilla fine-tuning and VPT approaches for the backbones pretrained using MAE and MoCo v3 on VTAB-1k. In the future, we will explore employing SPT to adapt large vision models to more downstream tasks, e.g., segmentation, detection, and pose estimation. Another promising direction is to further enhance the parameter and inference efficiency by simultaneously considering compressing these models. Finally, we will explore masking the gradients in the unstructured tuning of our SPT in a hardware-friendly way to save fine-tuning GPU memory consumption.

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

# Appendix

## A   MORE TASK-SPECIFIC SENSITIVITY PATTERNS

We show more patterns of block-wise task-specific sensitivity for tasks sampled from VTAB-1k (Zhai et al., 2019b) in Figures A, B, and C. We observe the similar trend that given a pre-trained backbone, the important parameters for different tasks diverse greatly. This aligns with our motivation to tune task-specific parameters.

## B   COMPARISON ON SENSITIVITY VARIANCES ACROSS THE PRE-TRAINING STRATEGIES

We further compare the variance of the task-specific sensitivities on VTAB-1k for backbones with different pre-training strategies. We present the results in Figure D. We observe that the backbones with self-supervised pre-training have a higher variance of sensitivities than the supervised pre-trained ones across the 19 downstream tasks. Especially, the variance for MAE (He et al., 2022b) is twice as large as the supervised pre-training strategy. We conjecture that our SPT yields higher performance than the other methods since it can better handle the large variance and tailor suitable trainable parameters for specific tasks.

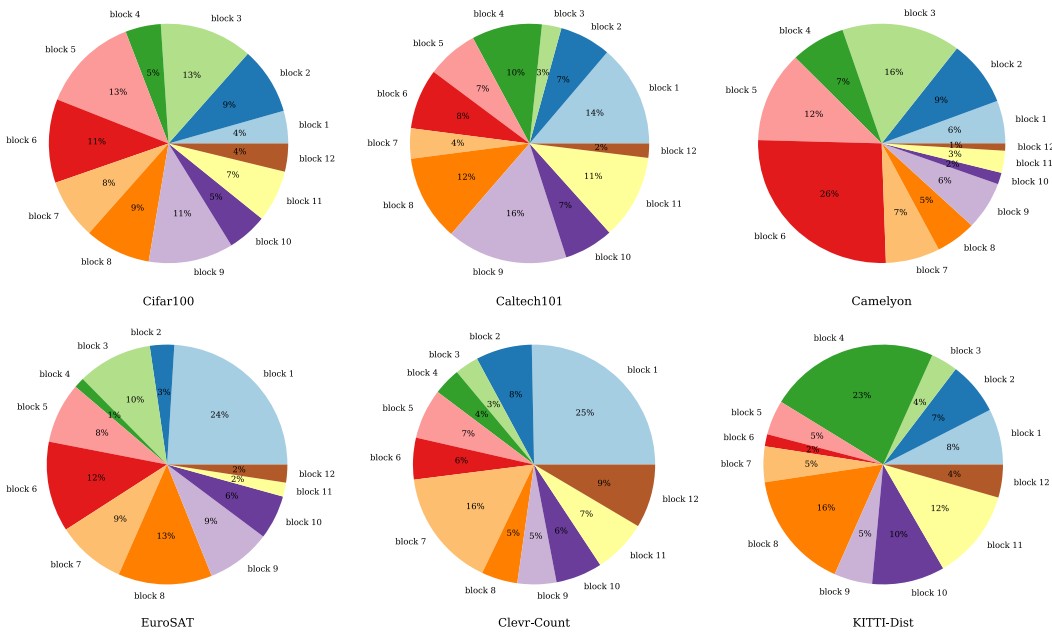

Figure A: The block-wise parameter sensitivity with ViT-B (Dosovitskiy et al., 2021) backbone under supervised pre-training for six sampled tasks from VTAB-1k.

## C   VISUALIZATION FOR DOMAIN GAPS

We visualize some sampled data for ImageNet, Natural tasks, and Structured tasks in Figure E to show the domain gaps. The domain gap between the pre-training data and the samples from Structured datasets is large as they mostly contain synthetic images. The visualization aligns with our finding that structured tasks have large domain gaps in the pre-training data which is handled by our SPT with structured tuning.

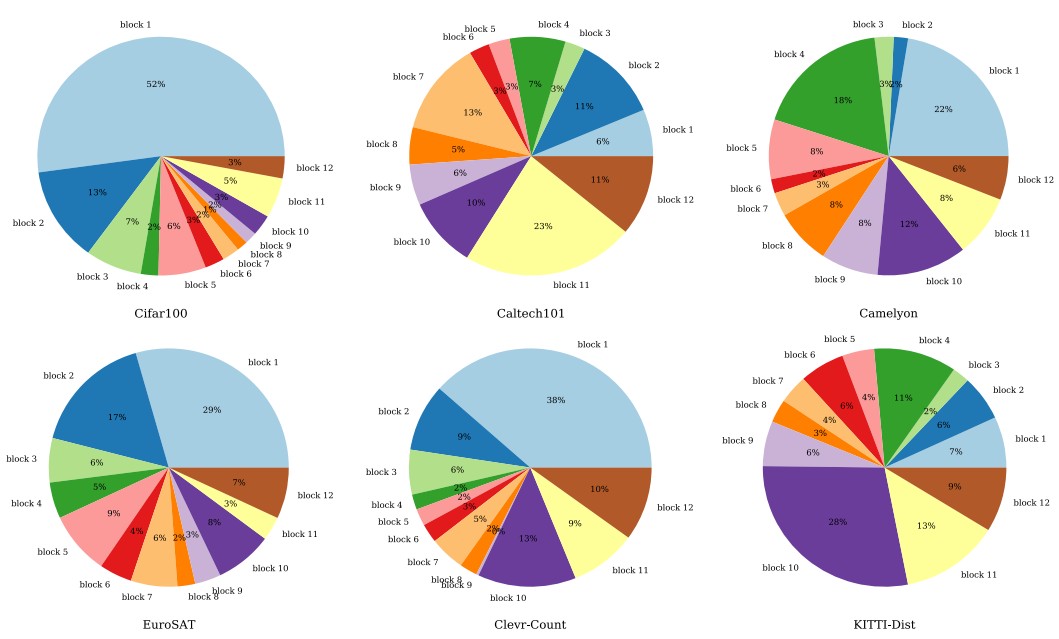

Figure B: The block-wise parameter sensitivity with ViT-B backbone under MAE pre-training for six sampled tasks from VTAB-1k.

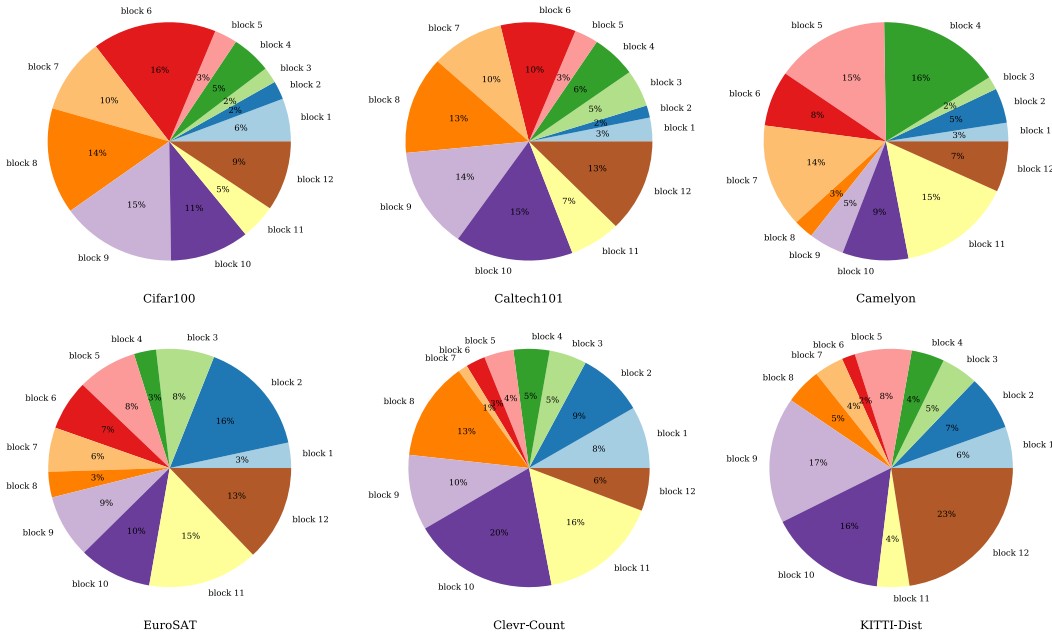

Figure C: The block-wise parameter sensitivity with ViT-B backbone under MoCo v3 (Chen et al., 2021) pre-training for six sampled tasks from VTAB-1k.

# D    SCALABILITY COMPARISON WITH LORA

We evaluate the scalability of our SPT and compare with LoRA. The results are shown in Figure D (b). We observe that LoRA saturates quickly while our SPT has consistent performance gain when the percentage of the trainable parameter increases. We conjecture that our SPT finds task-specific

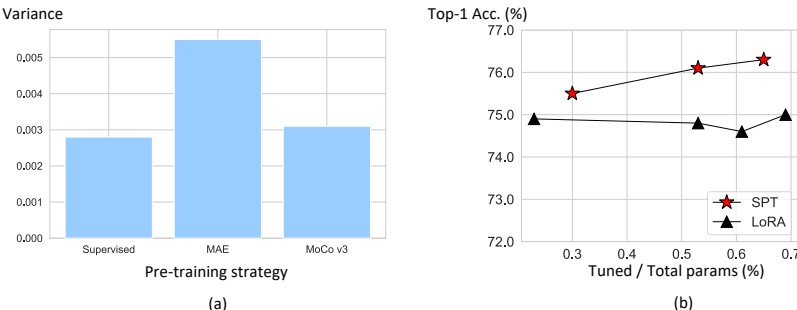

Figure D: (a) Comparison on the sensitivity variances across backbones with different pre-training strategies on VTAB-1k. (b) Comparison on the scalability between our SPT and LoRA (Hu et al., 2022) on VTAB-1k. LoRA saturates quickly while our SPT has consistent performance gain when the percentage of the trainable parameter increases.

Table A: Per-task results on the VTAB-1k benchmark from Table 1. "Tuned / Total" denotes the fraction of the trainable parameters. Top-1 accuracy (%) is reported.

| | Tuned / Total | Natural | | | | | | | | Specialized | | | | | Structured | | | | | | | | |
|---|---|---|---|---|---|---|---|---|---|---|---|---|---|---|---|---|---|---|---|---|---|---|---|
| | | CIfar100 | Caltech101 | DTD | Flower102 | Pets | SVHN | Sun397 | Mean Acc. | Camelyon | EuroSAT | Resisc45 | Retinopathy | Mean Acc. | Clevr-Count | Clevr-Dist | DMLab | KITTI-Dist | dSpr-Loc | dSpr-Ori | sNORB-Azim | sNORB-Ele | Mean Acc. |
| FULL | 100% | 68.9 | 87.7 | 64.3 | 97.2 | 86.9 | 87.4 | 38.8 | 75.9 | 79.7 | 95.7 | 84.2 | 73.9 | 83.4 | 56.3 | 58.6 | 41.7 | 65.5 | 57.5 | 46.7 | 25.7 | 29.1 | 47.6 |
| **Addition-based methods** | | | | | | | | | | | | | | | | | | | | | | | |
| MLP-3 | 1.50% | 63.8 | 84.7 | 62.3 | 97.4 | 84.7 | 32.5 | 49.2 | 67.8 | 77.0 | 88.0 | 70.2 | 56.1 | 72.8 | 47.8 | 32.8 | 32.3 | 58.1 | 12.9 | 21.2 | 15.2 | 24.8 | 30.6 |
| PROMPT-SHALLOW | 0.31% | 77.7 | 86.9 | 62.6 | 97.5 | 87.3 | 74.5 | 51.2 | 76.8 | 78.2 | 92.0 | 75.6 | 72.9 | 79.7 | 50.5 | 58.6 | 40.5 | 67.1 | 68.7 | 36.1 | 20.2 | 34.1 | 47.0 |
| PROMPT-DEEP | 0.98% | 78.8 | 90.8 | 65.8 | 98.0 | 88.3 | 78.1 | 49.6 | 78.5 | 81.8 | 96.1 | 83.4 | 68.4 | 82.4 | 68.5 | 60.0 | 46.5 | 72.8 | 73.6 | 47.9 | 32.9 | 37.8 | 55.0 |
| ADAPTER-8 | 0.39% | 69.2 | 90.1 | 68.0 | 98.8 | 89.9 | 82.8 | 54.3 | 79.0 | 84.0 | 94.9 | 81.9 | 75.5 | 84.1 | 80.9 | 65.3 | 48.6 | 78.3 | 74.8 | 48.5 | 29.9 | 41.6 | 58.5 |
| ADAPTER-32 | 0.71% | 68.7 | 92.2 | 69.8 | 98.9 | 90.3 | 84.2 | 53.0 | 79.6 | 83.2 | 95.4 | 83.2 | 74.3 | 84.0 | 81.9 | 63.9 | 48.7 | 80.6 | 76.2 | 47.6 | 30.8 | 36.4 | 58.3 |
| NOAH | 0.50% | 69.6 | 92.7 | 70.2 | 99.1 | 90.4 | 86.1 | 53.7 | 80.2 | 84.4 | 95.4 | 83.9 | 75.8 | 84.9 | 82.8 | 68.9 | 49.9 | 81.7 | 81.8 | 48.3 | 32.8 | 44.2 | 61.3 |
| **Reparameterization-based methods** | | | | | | | | | | | | | | | | | | | | | | | |
| LINEAR | 0.12% | 63.4 | 85.0 | 63.2 | 97.0 | 86.3 | 36.6 | 51.0 | 68.9 | 78.5 | 87.5 | 68.6 | 74.0 | 77.2 | 34.3 | 30.6 | 33.2 | 55.4 | 12.5 | 20.0 | 9.6 | 19.2 | 26.8 |
| PARTIAL-1 | 8.38% | 66.8 | 85.9 | 62.5 | 97.3 | 85.5 | 37.6 | 50.6 | 69.4 | 78.6 | 89.8 | 72.5 | 73.3 | 78.5 | 41.5 | 34.3 | 33.9 | 61.0 | 31.3 | 32.8 | 16.3 | 22.4 | 34.2 |
| BITFIT | 0.13% | 72.8 | 87.0 | 59.2 | 97.5 | 85.3 | 59.9 | 51.4 | 73.3 | 78.7 | 91.6 | 72.9 | 69.8 | 78.3 | 61.5 | 55.6 | 32.4 | 55.9 | 66.6 | 40.0 | 15.7 | 25.1 | 44.1 |
| LORA-8 | 0.55% | 67.1 | 91.4 | 69.4 | 98.8 | 90.4 | 85.3 | 54.0 | 79.5 | 84.9 | 95.3 | 84.4 | 73.6 | 84.6 | 82.9 | 69.2 | 49.8 | 78.5 | 75.7 | 47.1 | 31.0 | 44.0 | 60.5 |
| LORA-16 | 0.90% | 68.1 | 91.4 | 69.8 | 99.0 | 90.5 | 86.4 | 53.1 | 79.8 | 85.1 | 95.8 | 84.7 | 74.2 | 84.9 | 83.0 | 66.9 | 50.4 | 81.4 | 80.2 | 46.6 | 32.2 | 41.1 | 60.2 |
| SPT | 0.53% | 73.9 | 93.2 | 72.1 | 99.3 | 91.3 | 86.8 | 55.9 | 81.8 | 86.6 | 95.8 | 85.8 | 75.2 | 85.9 | 83.9 | 69.9 | 52.5 | 81.7 | 81.7 | 48.3 | 31.0 | 40.1 | 61.1 |
| Tuned / Total (%) | | 0.19% | 0.47% | 0.70% | 0.47% | 0.47% | 0.71% | 0.22% | 0.46% | 0.47% | 0.93% | 0.93% | 0.69% | 0.73% | 0.70% | 0.70% | 0.70% | 0.70% | 0.70% | 0.23% | 0.70% | 0.26% | 0.59% |
| SPT | 0.65% | 73.9 | 93.2 | 72.1 | 99.4 | 91.3 | 86.9 | 55.9 | 81.8 | 86.6 | 95.8 | 85.2 | 75.2 | 85.7 | 83.9 | 69.9 | 51.5 | 81.7 | 80.6 | 48.3 | 30.2 | 40.1 | 60.8 |
| Tuned / Total (%) | | 0.19% | 0.47% | 0.70% | 0.70% | 0.47% | 0.71% | 0.22% | 0.46% | 0.47% | 0.93% | 0.93% | 0.69% | 0.52% | 0.70% | 0.70% | 0.70% | 0.93% | 0.70% | 0.93% | 0.93% | 0.26% | 0.73% |

sensitive positions and adaptively combines unstructured and structured tuning granularities, thereby better allocating the trainable parameters than LoRA.

## E  HYPER-PARAMETERS AND AUGMENTATIONS.

We follow (Jia et al., 2022) for the hyper-parameter settings on MLP-$k$, PROMPT-SHALLOW, PROMPT-DEEP, PARTIAL-$k$, and BITFIT. For our SPT and the other methods, we use the AdamW optimizer (Loshchilov & Hutter, 2018) with cosine learning rate decay, batch size 64, learning rate $1 \times 10^{-3}$, and weight decay $1 \times 10^{-4}$ following (Zhang et al., 2022). We also follow (Zhang et al., 2022) to set the rank $r$ for structured tuning in SPT as 8. For the hyper-parameters specific to our SPT, we set the threshold $\sigma = 0.2$ in Eq. (4) by grid search on the validation set. The effect of varying $\sigma$ is analyzed in Section 4.3. Given the budget constraint $\tau$ on the number of trainable parameters (by default, less than 1%), we search for the optimal number since fewer parameters may have better performance. Therefore, we follow (Jia et al., 2022) and conduct grid search on the validation set. We search over {0.2M, 0.4M, 0.6M, 0.8M} and {0.2M, 0.4M, 0.6M, 0.8M, 1.0M} for VTAB-1k datasets and FGVC datasets, respectively. The full results and details for the searched number of trainable parameters for each task are presented in Table A and B of the appendix. We set the number of training samples $M$ used to calculate our parameter sensitivities to be 800 and 1,600 for VTAB-1k and FGVC datasets, respectively.

## F  EFFECT OF THE THRESHOLD HYPER-PARAMETER $\sigma$.

We investigate the effect of the threshold hyper-parameter $\sigma$ (in Eq. (4)). The results are shown in Table D under around 0.70% fractions of the trainable parameters. We see that when varying $\sigma$

Table B: Per-task results on the FGVC benchmark from Table 1. "Tuned / Total" denotes the fraction of the trainable parameters. Top-1 accuracy (%) is reported.

| | Tuned / Total | CUB-200-2011 | NABirds | Oxford Flowers | Stanford Dogs | Stanford Cars | Mean Acc. |
|---|---|---|---|---|---|---|---|
| FULL | 100% | 87.3 | 82.7 | 98.8 | 89.4 | 84.5 | 88.5 |
| **Addition-based methods** | | | | | | | |
| MLP-3 | 1.50% | 85.1 | 77.3 | 97.9 | 84.9 | 53.8 | 79.8 |
| PROMPT-SHALLOW | 0.31% | 86.7 | 78.8 | 98.4 | 90.7 | 68.7 | 84.6 |
| PROMPT-DEEP | 0.98% | 88.5 | 84.2 | 99.0 | 90.2 | 83.6 | 89.1 |
| ADAPTER-8 | 0.39% | 87.3 | 84.3 | 98.4 | 88.8 | 68.4 | 85.5 |
| ADAPTER-32 | 0.95% | 87.2 | 84.3 | 98.5 | 89.6 | 68.4 | 85.6 |
| **Reparameterization-based methods** | | | | | | | |
| LINEAR | 0.12% | 85.3 | 75.9 | 97.9 | 86.2 | 51.3 | 79.3 |
| PARTIAL-1 | 8.38% | 85.6 | 77.8 | 98.2 | 85.5 | 66.2 | 82.6 |
| BITFIT | 0.13% | 88.4 | 84.2 | 98.8 | 91.2 | 79.4 | 88.4 |
| LoRA-8 | 0.55% | 84.9 | 79.0 | 98.1 | 88.1 | 79.8 | 86.0 |
| LoRA-16 | 0.90% | 85.6 | 79.8 | 98.9 | 87.6 | 72.0 | 84.8 |
| SPT | 0.66% | 88.5 | 82.4 | 98.7 | 89.8 | 83.9 | 88.7 |
| Tuned / Total (%) | | 0.71% | 0.70% | 0.71% | 0.47% | 1.07% | 0.66% |
| SPT | 0.91% | 88.5 | 82.8 | 99.0 | 89.8 | 85.7 | 89.2 |
| Tuned / Total (%) | | 0.71% | 1.14% | 1.15% | 0.47% | 1.07% | 0.91% |

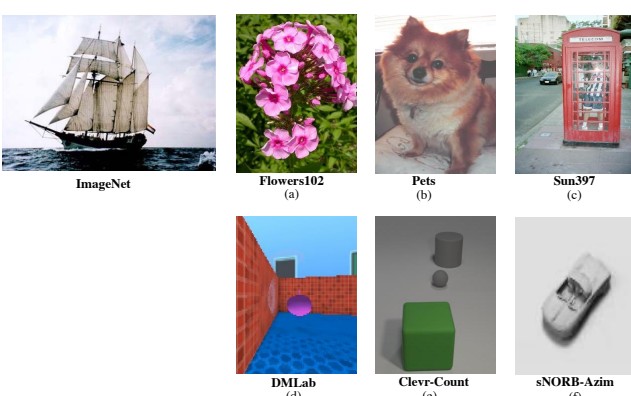

Figure E: Dataset samples from ImageNet, Natural tasks, and Structured tasks. We can observe that the samples from Natural tasks in VTAB-1k ((a), (b), and (c)) have clearly smaller domain gaps to the pre-training ImageNet (Krizhevsky et al., 2012) samples than the ones from Structured tasks in VTAB-1k ((d), (e), and (f)).

Table C: Multi-seed results for our SPT with ViT-B/16 under supervised, MAE (He et al., 2022b), and MoCo v3 (Chen et al., 2021) pre-training strategies. "Tuned / Total" denotes the fraction of the trainable parameters. The mean and standard deviations of the Top-1 accuracy (%) with three random seeds are reported.

| Method | Tuned / Total | Natural | Specialized | Structured | Mean Acc. |
|---|---|---|---|---|---|
| SPT, supervised | 0.62±0.04 | 81.77±0.17 | 85.67±0.06 | 61.01±0.20 | 76.20±0.06 |
| SPT, supervised | 0.54±0.04 | 81.65±0.11 | 85.85±0.07 | 60.75±0.09 | 76.02±0.07 |
| SPT, MAE | 0.54±0.04 | 62.10±0.29 | 79.02±0.57 | 57.87±0.66 | 66.32±0.41 |
| SPT, MAE | 0.42±0.02 | 61.95±0.45 | 78.72±0.50 | 57.23±0.53 | 65.97±0.28 |
| SPT, MoCo v3 | 0.56±0.06 | 72.26±0.45 | 84.38±0.25 | 58.58±0.20 | 71.74±0.22 |
| SPT, MoCo v3 | 0.48±0.07 | 71.01±0.45 | 83.51±0.45 | 56.89±0.44 | 70.47±0.10 |

from 0.05 to 0.40, the performance only differs for 0.1-0.2% mean top-1 accuracy. While setting $\sigma$ to 0.80 has a significant performance drop. As a higher $\sigma$ indicates employing more unstructured tuning, we conjecture that allocating too many trainable parameters to unstructured tuning limits the model's representational capability. As $\sigma = 0.2$ achieves the best performance, we consistently use $\sigma = 0.2$ for all experiments.

Table D: Ablate on the threshold hyper-parameter $\sigma$ for SPT on VTAB-1k. Top-1 accuracy (%) is reported.

| $\sigma$ | 0.05 | 0.10 | 0.20 | 0.40 | 0.80 |
|---|---|---|---|---|---|
| **Mean Acc.** | 76.1 | 76.2 | **76.3** | 76.2 | 73.9 |

