# OpenReview forum: "Sensitivity-aware Visual Parameter-efficient Tuning"
_ICLR.cc/2023/Conference — Submitted to ICLR 2023_

### Official Review · Reviewer_y14U · 2022-10-24

**Confidence:** 4
**Correctness:** 2
**Technical Novelty And Significance:** 3
**Empirical Novelty And Significance:** 3
**Recommendation:** 5

**Clarity, Quality, Novelty And Reproducibility:**

The paper clarified its contributions and approach most clearly.
In general, the paper is novel, can be reproduced and is of fair quality.


**Strength And Weaknesses:**

*[Strength]*
1. The motivation that focuses on task-driven parameter tuning is contributed.
2. This work proposed a novel criterion to efficiently measure the sensitivity (importance) of the pretrained backbone parameters to a specific task.

*[Weakness]*
1. The paper claimed there are challenges about lower inference speed, but in the experiments, there are no related experiments with other SOA methods in Table 1 (only comparisons with the PROMPT-DEEP in Table 6).
2. The experiments are about the ratio of the tuned parameter to the total parameter, instead of the comparison between the total tuned parameter number. There is a concern about the smaller ratio of tuned to total coming from more total parameters.
3. Does such a strategy lead to higher training time (fine-tuning time), compared with other SOA in table 1?
4. In section 3.2, how to get $P$ from $S$?



**Summary Of The Paper:**

This work mainly focuses on the lower inference speed and lacking effective configurations for trainable parameters tailored for each task. It proposed a simple but effective approach named Sensitivity-aware visual Parameter-efficient Tuning (SPT) for these challenges. It quickly identifies the important parameters of the given task in a data-dependent way before fine-tuning without the complex selection schedule. Also, low-rank reparameterization is employed to achieve a better trade-off between efficiency and accuracy. The experiments are conducted on 24 downstream tasks.

**Summary Of The Review:**

This paper proposed a novel criterion to efficiently measure the sensitivity (importance) of the pretrained backbone parameters, and proposed a parameter-efficient tuning approach SPT to tune parameters at task-specific important positions.
However, there are a few concerns about if the experiments validate the claimed contributions.

---

> ### Author Response · Authors · 2022-11-17
> **Response to Reviewer y14U**
>
> Thanks for your constructive comments.
>
> **Q1**: Why not compare the inference speed with SOTA methods?
>
> **A1**: We have already described in Section 2 that the SOTA methods can be categorized into addition-based methods and reparameterization-based methods. For the addition-based methods, "the additional parameters require extra computations compared to the full fine-tuning, thereby introducing additional lags during inference", while "Reparameterization-based methods tune the parameters that are inherently in the backbone or new parameters that can be merged into the backbone, thereby yielding no extra computational costs during inference." As our SPT belongs to Reparameterization-based methods, it has no inference lag compared to the full fine-tuning. To fully demonstrate the inference lag differences between the two categories, we have reported more inferences speed in Table 1 of the revised manuscript.
>
> **Q2**: The concern for the total number of trainable parameters.
>
> **A2**: We have added the total number of trainable parameters with the blue color in the revised manuscript in Tables 1 and 2 and observed that the two metrics are similar in measuring the number of trainable parameters for our SPT.
>
> **Q3**: The concern for fine-tuning time.
>
> **A3**: Our strategy only leads to marginally higher training time. Specifically, our SPT needs an additional stage ahead of the training to calculate the parameter sensitivity, which is around 5.5 seconds with a batch size of 64 on a single V100 GPU averaged for the 19 datasets for VTAB-1k benchmark. Then, training our SPT for 100 epochs requires slightly higher training time than LoRA: training on VTAB-1k benchmark takes around 970 seconds for LoRA and around 1,059 seconds for our SPT averaged over the 19 datasets. The slightly higher training time is used by masking the gradients for unstructured tuning.  However, we argue that **the primary aim of this paper is to tune parameters at task-specific important positions while being inference-efficient instead of saving the fine-tuning time**. Please refer to more details in Response to Reviewer MqQs, Q2.
>
> **Q4**: How to construct the $\mathcal{P}$ which stores indexes for the sensitive connections within $\boldsymbol{W}$?
>
> **A4**: $\mathcal{P}$ (in the previous version) stores the indexes of the sensitive parameters for any weight matrix. We have revised the related parts to describe our approach better and marked them in blue. Specifically, we first derive the top-$\tau$ sensitive parameters in set $\mathcal{T}$ according to their sensitivity in $\mathcal{S}$. Then, in Eq. (3), we introduce a binary mask for each weight matrix that is the same size as the matrix. The non-zero indexes in the binary mask ($\mathcal{P}$ in the previous version) indicate the sensitive connections in $\mathcal{T}$ that are within the weight matrix. We finally mask the gradients with the binary mask to conduct unstructured tuning.

---

> ### Author Response · Authors · 2022-11-22
> **Request for Discussion**
>
> Dear Reviewer y14U,
>
> We sincerely thank you again for your great efforts in reviewing this paper. We have addressed your concerns regarding the experimental settings, including total tuned parameters, inference time, and fine-tuning time. Please don’t hesitate to let us know if there are still concerns/questions.
>
> Best regards,
>
> Authors of #1187

---

> ### Author Response · Authors · 2022-12-10
> **Keenly looking forward to your post-rebuttal feedback**
>
> Dear Reviewer y14U,
>
> Thanks again for your time in reviewing our work. As the rebuttal discussion is about to end soon, would you mind checking our responses ([a short summary](https://openreview.net/forum?id=9GOjmbRQ2o&noteId=zkh9AXP-H5) and the point-to-point answers) and confirming whether you have any further questions?
>
> If you have any more questions, we are very happy to discuss!
>
>
> Best regards,
>
> Authors of #1187

---

> ### Author Response · Authors · 2022-12-12
> **Final Call for Your Feedback**
>
> Dear Reviewer y14U,
>
> Thanks again for your engagement in reviewing our work. Since we are nearly at the end of the discussion phase, we sincerely hope you can consider positively recommending our work if your concerns are solved. If you still have further comments/suggestions, please don't hesitate to let us know.
>
> Best regards,
>
> Authors of #1187

---

### Official Review · Reviewer_Z771 · 2022-10-25

**Confidence:** 3
**Correctness:** 2
**Technical Novelty And Significance:** 2
**Empirical Novelty And Significance:** 2
**Recommendation:** 5

**Clarity, Quality, Novelty And Reproducibility:**

The idea is novel but the clarity is a concern, hard to reproduce if the paper does not release code

**Strength And Weaknesses:**

Strengths: (1) a simple way to identify sensitive visual parameters.
(2) a low-rank reparameterization approach to update the visual parameters
(3) good results on FGVC and VTAB datasets


Weakness: (1) The introduction on sensitive visual parameter identification is kind of verbal. The criteria is simply the first order gradient and  it would not be efficient to compute second-order hessian matrix. Introducing eq 2 and 3 would be helpful because you can not find optimal in anyways.
(2) Missing introduction of how to compute W_up and W_down


**Summary Of The Paper:**

This paper proposes to first recognize influential visual parameters by its averaged gradients and then tune those parameters. In order to improve adaptation capability, the paper also propose to switch to a low-rank reparameterization of the weight matrix that contains enough sensitive parameters. The paper evaluated on both FGVC and VTAB dataset and achieve even better results compared to fully fine-tuned model due to the small size of the fine-tuned dataset.

**Summary Of The Review:**

The paper propose a pipeline for recognizing and tuning visual parameters. The idea is interesting but my concern is on the clarity part

---

> ### Author Response · Authors · 2022-11-17
> **Response to Reviewer Z771**
>
> Thanks for your constructive comments.
>
> **Q1**: Would it be inefficient to compute a higher-order Taylor expansion?
>
> **A1**: We authors would like to clarify that we have already analyzed the differences between first-order and higher-order Taylor expansions in Section 3.1 in the initial submission. A higher order Taylor expansion only makes slight differences but a higher computational cost. Therefore, the first-order expansion is a fast and accurate approximation, and we do not need higher-order expansions.
>
> **Q2**: Missing introduction of how to compute $\boldsymbol{W}\_{\rm up}$ and $\boldsymbol{W}\_{\rm down}$.
>
> **A2**: Thanks for pointing this out. $\boldsymbol{W}\_{\rm up}$ and $\boldsymbol{W}\_{\rm down}$ are randomly initialized learnable parameters. We have revised the related parts and marked them in blue.
>
> **Q3**: Reproducibility concern.
>
> **A3**: We have included the code to reproduce the results in the [supplementary material](https://openreview.net/attachment?id=9GOjmbRQ2o&name=supplementary_material). We will also release the code to the public after acceptance.

---

> ### Author Response · Authors · 2022-11-22
> **Request for Discussion**
>
> Dear Reviewer Z771,
>
> We sincerely thank you again for your great efforts in reviewing this paper. We have addressed your concerns regarding the reproducibility and clarity of the paper (including code in the supplementary material and improving the presentation of our paper in the revised manuscript). Please don’t hesitate to let us know if there are still concerns/questions.
>
> Best regards,
>
> Authors of #1187

---

> ### Author Response · Authors · 2022-12-10
> **Keenly looking forward to your post-rebuttal feedback**
>
> Dear Reviewer Z771,
>
> Thanks again for your time in reviewing our work. As the rebuttal discussion is about to end soon, would you mind checking our responses ([a short summary](https://openreview.net/forum?id=9GOjmbRQ2o&noteId=zkh9AXP-H5) and the point-to-point answers) and confirming whether you have any further questions?
>
> If you have any more questions, we are very happy to discuss!
>
>
> Best regards,
>
> Authors of #1187

---

> ### Author Response · Authors · 2022-12-12
> **Final Call for Your Feedback**
>
> Dear Reviewer Z771,
>
> Thanks again for your engagement in reviewing our work. Since we are nearly at the end of the discussion phase, we sincerely hope you can consider positively recommending our work if your concerns are solved. If you still have further comments/suggestions, please don't hesitate to let us know.
>
> Best regards,
>
> Authors of #1187

---

### Official Review · Reviewer_hvdx · 2022-10-25

**Confidence:** 5
**Correctness:** 4
**Technical Novelty And Significance:** 3
**Empirical Novelty And Significance:** 3
**Recommendation:** 6

**Clarity, Quality, Novelty And Reproducibility:**

- Clarity and Quality: the paper is well-written and easy to understand, while there is only one part I would suggest the authors further clarify.

- Novelty: the proposed structured and unstructured tuning is new to the parameter-efficient method to the best of my knowledge.

- Reproducibility: the authors seem not to provide the code implementation, but it might be reproducible given the provided implementation details.

**Strength And Weaknesses:**

**Strength**
- The paper is well written, and most of the proposed components are easy to understand. The figures with a clear explanation help the reader follow the idea.

- The proposed idea is novel and can potentially adapt to different datasets with task-wise numbers of trainable parameters. Applying structured and unstructured tuning in parameter-efficient learning is new to the community to the best of my knowledge.

- The authors show extensive results on different datasets and pretrained models, and the following analyses confirm the effectiveness of the proposed method.

**Weakness**
- If I understand correctly, one potential weakness is the computation time of two-stage training. The proposed SPT requires an additional stage to estimate the sensitivity of each parameter based on the gradient and decide which parameters are decomposed into low-rank matrices. Then the proposed method tunes these chosen parameters to adapt to new tasks. This might make the computation time longer than simple one-stage training methods.

- For $\Delta_w$ in equation 6, it is confusing when the authors explain  $\Delta_w$ means in the following paragraphs. First $(i,j) \notin P$ is confused (it might mean that these parameters are not in the selected parameters set, but this notation is quite confusing). My guess for what the authors try to express in the second line of Eq. 6 is that the unstructured tuning is applied on these weight matrices where only a few parameters need to be tuned. I would suggest the authors clarify this part.

- I would suggest the authors report the variance of reported results since the parameter-efficient method might be sensitive to the randomness in the training.


**Summary Of The Paper:**

This paper aims to perform parameter-efficient adaption for vision tasks, where only a small number of parameters are trained to adapt to new tasks (image classification tasks in this paper). To address this problem, the authors propose to exploit the estimated sensitivity to decide which parameters to be tuned, and the sensitivity is measured according to the gradients. Based on the ratio of sensitive parameters in the weight matrix, the authors use structured or unstructured tuning. In the experiment, the authors show that their proposed SPT can perform better using fewer parameters than the previous parameter-efficient methods in VTAB-1k and FGVC experiments. Additionally, they show some analyses, including using self-supervised pretrained backbones, using or not using structured tuning, and other hyper-parameter tunings.



**Summary Of The Review:**

This paper provides a new parameter-efficient method for visual tasks, and they provide extensive results on different datasets and show the improvements against prior works. Given the current status of the paper, I would recommend the score "marginally above the acceptance threshold".

---

> ### Author Response · Authors · 2022-11-17
> **Response to Reviewer hvdx**
>
> Thanks for your constructive comments.
>
> **Q1**: Does SPT consume much more computation time due to calculating sensitivity?
>
> **A1**: Actually, our SPT only requires slightly more time in the first stage. As shown in Table 5, averaging the gradient of only half of the training samples (0.5 epoch) is enough to get the best results. Since the training takes a total of 100 epochs, the first stage only takes 0.5% of the original training time, which is around 5.5 seconds, with a batch size of 64 on a single V100 GPU averaged for the 19 VTAB-1k datasets.
>
> **Q2**: Clarification on Eq. (6).
>
> **A2**: Thanks for pointing this out. We have revised the related parts (Section 3.2) and marked them in blue. Specifically, in Eq. (3), we introduce a binary mask for each weight matrix that is the same size as the matrix. The non-zero indexes in the binary mask indicate the sensitive connections within the weight matrix. We finally mask the gradients with the binary mask to conduct unstructured tuning.
>
> **Q3**: Variance of reported results.
>
> **A3**: Thanks for your suggestions. We report the multi-seed main results for SPT with three random seeds in Table J.
>
> Table J. Multi-seed results with ViT-B/16. ``Tuned / Total'' denotes the fraction of the trainable parameters. Top-1 accuracy (\%) is reported.
>
> | **Method**      | **Tuned / Total** | Natural        | **Specialized** | **Structured** | **Mean**       |
> | --------------- | ----------------- | -------------- | --------------- | -------------- | -------------- |
> | SPT, supervised | 0.62$\pm$0.04     | 81.77$\pm$0.17 | 85.67$\pm$0.06  | 61.01$\pm$0.20 | 76.20$\pm$0.06 |
> | SPT, supervised | 0.54$\pm$0.04     | 81.65$\pm$0.11 | 85.85$\pm$0.07  | 60.75$\pm$0.09 | 76.02$\pm$0.07 |
> | SPT, MAE        | 0.54$\pm$0.04     | 62.10$\pm$0.29 | 79.02$\pm$0.57  | 57.87$\pm$0.66 | 66.32$\pm$0.41 |
> | SPT, MAE        | 0.42$\pm$0.02     | 61.95$\pm$0.45 | 78.72$\pm$0.50  | 57.23$\pm$0.53 | 65.97$\pm$0.28 |
> | SPT, MoCo v3    | 0.56$\pm$0.06     | 72.26$\pm$0.45 | 84.38$\pm$0.25  | 58.58$\pm$0.20 | 71.74$\pm$0.22 |
> | SPT, MoCo v3    | 0.48$\pm$0.07     | 71.01$\pm$0.45 | 83.51$\pm$0.45  | 56.89$\pm$0.44 | 70.47$\pm$0.10 |
>
> We observe that our SPT still outperforms the SOTA methods by large margins. We also observe that the standard deviation is slightly higher for SPT under MAE and MoCo v3 pre-training strategies, which is interesting for future investigation. We have also included these results in Table C of the Appendix in the revised manuscript.

---

> ### Author Response · Authors · 2022-11-22
> **Request for Discussion**
>
> Dear Reviewer hvdx,
>
> We sincerely thank you again for your great efforts in reviewing this paper. We have addressed your concerns regarding the clarity of the paper and the variance of the reported results for our SPT. Please don’t hesitate to let us know if there are still concerns/questions.
>
> Best regards,
>
> Authors of #1187

---

> ### Author Response · Authors · 2022-12-10
> **Keenly looking forward to your post-rebuttal feedback**
>
> Dear Reviewer hvdx,
>
> Thanks again for your time in reviewing our work. As the rebuttal discussion is about to end soon, would you mind checking our responses ([a short summary](https://openreview.net/forum?id=9GOjmbRQ2o&noteId=zkh9AXP-H5) and the point-to-point answers) and confirming whether you have any further questions?
>
> If you have any more questions, we are very happy to discuss!
>
>
> Best regards,
>
> Authors of #1187

---

> ### Author Response · Authors · 2022-12-12
> **Final Call for Your Feedback**
>
> Dear Reviewer hvdx,
>
> Thanks again for your engagement in reviewing our work. Since we are nearly at the end of the discussion phase, we sincerely hope you can consider positively recommending our work if your concerns are solved. If you still have further comments/suggestions, please don't hesitate to let us know.
>
> Best regards,
>
> Authors of #1187

---

### Official Review · Reviewer_MqQs · 2022-10-25

**Confidence:** 4
**Correctness:** 4
**Technical Novelty And Significance:** 2
**Empirical Novelty And Significance:** 2
**Recommendation:** 3

**Clarity, Quality, Novelty And Reproducibility:**

The clarity and reproducibility are generally acceptable. I am not satisfied with the novelty because the proposed method is neither theoretically insightful nor practically useful. Please see my comments above.

**Strength And Weaknesses:**

Strengths

S1. The idea is simple and easy to follow.

S2. The experimental results are acceptable.

Weaknesses

W1. The writing can be significantly improved. It is difficult to get what is new and what is significant in this paper. For example, after mentioning the criterion (Page 2), the authors commented finetuning only the important parameters (referred to as unstructured tuning) is not enough. The authors proposed to use LoRA (referred to as structured tuning). In other words, the proposed method only works in some cases and should be combined with LoRA. This significantly undermines the proposed method.

W2. The proposed method actually increases memory usage (Table 6), so why is that called efficient finetuning?

W3. As seen in Table 2, the proposed method is not impressively effective.

**Summary Of The Paper:**

This paper assumes that not all parameters contribute equally to the performance across different tasks. This paper proposes a criterion to measure the sensitivity of parameters, which is the magnitude of the gradient. Reasonable results are reported.

**Summary Of The Review:**

I rate it a rejection because the novelty is not satisfying. To me, the proposed method seems yet another LoRA with trivial modifications (i.e., in some cases we should select some parameters and finetune them in an unstructured way).

---

> ### Author Response · Authors · 2022-11-17
> **Response to Reviewer MqQs**
>
> Thanks for your constructive comments.
>
> **Q1**: What’s our novelty over LoRA [A]? Does our SPT only work in some cases?
>
> **A1**: We authors would like to clarify that our SPT is **fundamentally different from LoRA** (Major Q1) and is **orthogonal to LoRA** (Major Q2) as well as other VPT methods. Our SPT also does not "only work in some cases", as we adaptively determine the tuning granularities by Eq. (4) (In the revised manuscript) for each weight matrix in the pre-trained backbone in all cases and have demonstrated better performance (Tables 1 and 2).
>
> **Q2**: The proposed method actually increases memory usage (Table 6 in the previous version), so why is that called efficient finetuning?
>
> **A2**: As defined in the literature, the concept of parameter-efficient tuning is to alleviate the storage burden for deploying large models to different tasks, e.g., "Using GPT-3 175B as an example – deploying independent instances of fine-tuned models, each with 175B parameters, is prohibitively expensive." (in [A]); "...requires one to store and deploy a separate copy of the backbone parameters for every single task. This is an expensive and often infeasible proposition." (in [D]) and "...this results in a separate copy of fine-tuned model parameters for each task, which is prohibitively expensive when serving models that perform a large number of tasks." (in [B]). Also, compared to the pre-training, which takes thousands of TPUv3-core-days [G], the fine-tuning process for the total 19 datasets of the VTAB-1k benchmark only takes several GPU hours and is computationally friendly. We argue that the primary purpose of parameter-efficient tuning is saving the number of trainable parameters instead of reducing the memory consumption during fine-tuning and the marginally higher fine-tuning memory (0.6 GB) than the full fine-tuning for our SPT is affordable. The higher GPU memory usage for our SPT is caused by masking out the gradient for the less sensitive parameters. We take improving our SPT to be more hardware-friendly in terms of fine-tuning memory as future work.
>
> **Q3**: As seen in Table 2, the proposed method is not impressively effective.
>
> **A3**: We would like to highlight that SPT achieves **overall better performance than the SOTA methods** in Table 2. Specifically, SPT outperforms the second-best method (LoRA-16) by 1.3% mean accuracy with less trainable parameters using MAE pre-trained ViT-B. SPT also outperforms the second-best method (full fine-tuning) by 1.4% mean accuracy with only 0.51% trainable parameters using MoCo v3 pre-trained ViT-B. The improvements are non-trivial as praised by Reviewers Sx5t and Z771: "The results are strong in comparison." (Reviewer Sx5t). "Good results on FGVC and VTAB datasets." (reviewer Z771).
>
> Besides, as far as we know, SPT is also the **FIRST** VPT approach to consistently outperform the full fine-tuning on the two backbones with self-supervised pre-training strategies: surpassing the full fine-tuning by 1.8% and 1.9% mean accuracy for MAE [E] and MoCo v3 [F] pre-trained ViT-B/16. We argue that the mean accuracy gains that are close to 2.0% with only around 0.5% trainable parameters of full fine-tuning are non-trivial.
>
> **Q4**: The writing needs to be improved.
>
> **A4**: Thanks for your valuable suggestion. Following your suggestion, we have largely revised the manuscript in blue color, including emphasizing the significance of the paper (Section 1), describing our method much clearer (Section 3.2, Figures 1 (b) and 2), and introducing the discussion between our SPT and LoRA [A] (Section 3.2). We leave the rest of the manuscript intact, as praised by the other reviewers: "The writing is reasonably clear and fine to understand" (Reviewer Sx5t). "The paper is well written, and most of the proposed components are easy to understand. The figures with a clear explanation help the reader follow the idea." and "The paper is well-written and easy to understand." (Reviewer hvdx).

---

> > ### Author Response · Authors · 2022-11-17
> > **Response to Reviewer MqQs (Part 2)**
> >
> > **Reference**
> >
> > [A] Edward J Hu, yelong shen, Phillip Wallis, Zeyuan Allen-Zhu, Yuanzhi Li, Shean Wang, Lu Wang, and Weizhu Chen. LoRA: Low-rank adaptation of large language models. In ICLR, 2022.
> >
> > [B] Menglin Jia, Luming Tang, Bor-Chun Chen, Claire Cardie, Serge Belongie, Bharath Hariharan, and Ser-Nam Lim. Visual prompt tuning. In ECCV, 2022.
> >
> > [C] Neil Houlsby, Andrei Giurgiu, Stanislaw Jastrzebski, Bruna Morrone, Quentin De Laroussilhe, Andrea Gesmundo, Mona Attariyan, and Sylvain Gelly. Parameter-efficient transfer learning for nlp. In ICML, pp. 2790–2799, 2019.
> >
> > [D] Junxian He, Chunting Zhou, Xuezhe Ma, Taylor Berg-Kirkpatrick, and Graham Neubig. Towards a unified view of parameter-efficient transfer learning. In ICLR, 2022a.
> >
> > [E] Kaiming He, Xinlei Chen, Saining Xie, Yanghao Li, Piotr Dollar, and Ross Girshick. Masked autoencoders are scalable vision learners. In CVPR, pp. 16000–16009, 2022b.
> >
> > [F] Xinlei Chen, Saining Xie, and Kaiming He. An empirical study of training self-supervised vision transformers. In ICCV, pp. 9640–9649, 2021.
> >
> > [G] Alexey Dosovitskiy, Lucas Beyer, Alexander Kolesnikov, Dirk Weissenborn, Xiaohua Zhai, Thomas Unterthiner, Mostafa Dehghani, Matthias Minderer, Georg Heigold, Sylvain Gelly, Jakob Uszkoreit, and Neil Houlsby. An image is worth 16x16 words: Transformers for image recognition at scale. In ICLR, 2021.

---

> ### Author Response · Authors · 2022-11-22
> **Request for Discussion**
>
> Dear Reviewer MqQs,
>
> We sincerely thank you again for your great efforts in reviewing this paper. We have addressed your major concerns regarding the novelty and significance of the paper over LoRA. We have also improved the presentation in Abstract, Sections 1 and 3.2, Figures 1 and 2 to highlight the significance of our paper in the revised submission. Please don’t hesitate to let us know if there are still concerns/questions.
>
> Best regards,
>
> Authors of #1187

---

> ### Author Response · Authors · 2022-12-10
> **Keenly looking forward to your post-rebuttal feedback**
>
> Dear Reviewer MqQs,
>
> Thanks again for your time in reviewing our work. As the rebuttal discussion is about to end soon, would you mind checking our responses ([a short summary](https://openreview.net/forum?id=9GOjmbRQ2o&noteId=zkh9AXP-H5) and the point-to-point answers) and confirming whether you have any further questions?
>
> If you have any more questions, we are very happy to discuss!
>
>
> Best regards,
>
> Authors of #1187

---

> ### Author Response · Authors · 2022-12-12
> **Final Call for Your Feedback**
>
> Dear Reviewer MqQs,
>
> Thanks again for your engagement in reviewing our work. Since we are nearly at the end of the discussion phase, we sincerely hope you can consider positively recommending our work if your concerns are solved. If you still have further comments/suggestions, please don't hesitate to let us know.
>
> Best regards,
>
> Authors of #1187

---

### Official Review · Reviewer_Sx5t · 2022-10-25

**Confidence:** 4
**Correctness:** 3
**Technical Novelty And Significance:** 3
**Empirical Novelty And Significance:** 2
**Recommendation:** 5

**Clarity, Quality, Novelty And Reproducibility:**

The writing is reasonably clear and fine to understand.

The novelty is some limited as the new bit of the whole model is identification and update of top most sensitive parameters, which needs to base on LoRA to work. As a result, the overall quality is some less conniving.

The details given should be good for reproducibility.

**Strength And Weaknesses:**

**Strength**

- The problem of VPT efficiency is valid and valuable to be solved

- The idea of measuring the parameter sensitivity using network pruning methods is sensible and also novel in the VPT context. Technically, using the first derivative of Taylor expansion is a good choice with efficiency

- The results are strong in comparison.

**Weakness**

- It is some strange that the proposed method still update the whole parameters when a high proportion of parameters are sensitive to a specific task.

- Method: It is largely similar to LoRA except an introduction of a threshold based choice /branch. SPT w/o unstructured is equivalent to LoRA. So it is very incremental over LoRA.

- Experiment: Table 6 lacks comparison to LoRA.


**Summary Of The Paper:**

This paper considers the efficiency issue with visual parameter-efficient tuning (VPT), such as prompt learning and adapters. In particular, the idea is to tune only those important parameters of a pre-trained model per task, under an assumption that the importance of parameters is task specific. The proposed method is based on identifying the sensitivity of each parameters w.r.t a specific task based on previous model pruning methods. For performance, the whole parameters are still updated subject to a low-rank constraint for parameter efficiency, and further merge them into the backbone after fine-tuning.


**Summary Of The Review:**

This paper aims to adapt a pertained model to downstream tasks at no additional inference cost. The key idea is to find out what are top sensitive parameters for a given task, on which the update should be focused on. Previous pruning methods can be used to measure such sensitivity, and the final model needs to combine with LoRA according to some threshold. The overall method is some incremental over LoRA.

---

> ### Author Response · Authors · 2022-11-17
> **Response to Reviewer Sx5t**
>
> Thanks for your constructive comments.
>
> **Q1**: Is our SPT incremental over LoRA [A]?
>
> **A1**: Please refer to Major Q1. To summarize, **our paper is NOT simply LoRA with threshold**, as our SPT 1) identifies the task-specific sensitive positions, instead of heuristic positions like the other VPT approaches (LoRA [A], Prompt [B], and Adapter [C]) to tune the trainable parameters; 2) adaptively combines the unstructured tuning (tuning the weight connections) and structured tuning (tuning low-rank reparameterizations of updating the entire weight matrices) granularities to achieve higher flexibility and stronger representational capability simultaneously. The novelty of our SPT has also been praised by Reviewers hvdx, Z771, and y14U.
>
> **Q2**: Does our SPT have to base on LoRA to work?
>
> **A2**: **Our SPT is orthogonal to LoRA** and does not have to base on LoRA to work. Please refer to Major Q2.
>
> **Q3**: Why update the entire weight matrices when a high proportion of parameters is sensitive to a specific task?
>
> **A3**: The reason for updating the entire weight matrix with structured tuning has already been explained in the initial submission "...update the entire weight matrix to strengthen the adaptation ability…" (Section 3.2). Then, "to preserve the parameter budget when tuning one entire weight matrix, we freeze the pre-trained weights and reparameterize the update with two trainable low-rank weight matrices in a parameter-efficient way as in LoRA (Hu et al., 2022)." (Section 3.2). We have also validated our design choice in Section 4.3, Figure 3 (a), and Table 3 to show that structured tuning with higher representational capability facilitates handling the downstream tasks with a large domain gap from the pre-training data.
>
> **Q4**: More cost comparisons to LoRA.
> We have revised Table 5 (Table 6 in the previous version) and included more cost comparisons.
>
> Table 5. Cost comparison on VTAB-1k. The methods are evaluated with around 0.70\% fractions of the trainable parameters. We report the latency (ms/img) and the peak memory usage (GB).
> | **Method**           | **Inference Latency (ms/img)** | **Inference Memory (GB)** | **Fine-tuning Memory (GB)** |
> | -------------------- | ------------------------------ | ------------------------- | ------------------------- |
> | FULL                 | 2.8                            | 1.3                       | 11.9                      |
> | PROMPT-DEEP          | 3.8                            | 1.9                       | 13.2                      |
> | LoRA                 | 2.8                            | 1.3                       | 8.2                       |
> | SPT w/o unstructured | 2.8                            | 1.3                       | 8.3                       |
> | SPT                  | 2.8                            | 1.3                       | 12.5                      |
>
> **A4**: We argue that **the primary purpose of parameter-efficient tuning is saving the number of trainable parameters instead of reducing the memory consumption during fine-tuning**. As defined in the literature, the concept of parameter-efficient tuning is to alleviate the storage burden for deploying large models to different tasks, e.g., "Using GPT-3 175B as an example – deploying independent instances of fine-tuned models, each with 175B parameters, is prohibitively expensive." (in [A]); "...requires one to store and deploy a separate copy of the backbone parameters for every single task. This is an expensive and often infeasible proposition." (in [D]) and "...this results in a separate copy of fine-tuned model parameters for each task, which is prohibitively expensive when serving models that perform a large number of tasks." (in [B]).
>
> Our SPT is a reparameterization-based method, thereby having no extra inference latency and memory than the full fine-tuning. However, it is noteworthy that our unstructured tuning is implemented by masking out the gradient for the less sensitive parameters and thus has slightly higher fine-tuning memory consumption (0.6 GB higher than the full fine-tuning). Compared to the pre-training, which takes thousands of TPUv3-core-days [E], the fine-tuning process for the total 19 datasets of the VTAB-1k benchmark only takes several GPU hours and is computationally friendly. Therefore, we argue that slightly more fine-tuning memory for our SPT is affordable and saving fine-tuning memory is not the primary purpose of parameter-efficient tuning. Nevertheless, our SPT w/o unstructured tuning has similar fine-tuning memory as LoRA but higher performance (Figure 3 (b)). We take improving our SPT to be more hardware-friendly in terms of fine-tuning memory as future work.

---

> > ### Author Response · Authors · 2022-11-17
> > **Response to Reviewer Sx5t (Part 2)**
> >
> > **Reference**
> >
> > [A] Edward J Hu, yelong shen, Phillip Wallis, Zeyuan Allen-Zhu, Yuanzhi Li, Shean Wang, Lu Wang, and Weizhu Chen. LoRA: Low-rank adaptation of large language models. In ICLR, 2022.
> >
> > [B] Menglin Jia, Luming Tang, Bor-Chun Chen, Claire Cardie, Serge Belongie, Bharath Hariharan, and Ser-Nam Lim. Visual prompt tuning. In ECCV, 2022.
> >
> > [C] Neil Houlsby, Andrei Giurgiu, Stanislaw Jastrzebski, Bruna Morrone, Quentin De Laroussilhe, Andrea Gesmundo, Mona Attariyan, and Sylvain Gelly. Parameter-efficient transfer learning for nlp. In ICML, pp. 2790–2799, 2019.
> >
> > [D] Junxian He, Chunting Zhou, Xuezhe Ma, Taylor Berg-Kirkpatrick, and Graham Neubig. Towards a unified view of parameter-efficient transfer learning. In ICLR, 2022a.
> >
> > [E] Alexey Dosovitskiy, Lucas Beyer, Alexander Kolesnikov, Dirk Weissenborn, Xiaohua Zhai, Thomas Unterthiner, Mostafa Dehghani, Matthias Minderer, Georg Heigold, Sylvain Gelly, Jakob Uszkoreit, and Neil Houlsby. An image is worth 16x16 words: Transformers for image recognition at scale. In ICLR, 2021.

---

> ### Author Response · Authors · 2022-11-22
> **Request for Discussion**
>
> Dear Reviewer Sx5t,
>
> We sincerely thank you again for your great efforts in reviewing this paper. We have addressed your major concerns regarding the novelty and significance of the paper over LoRA. Please don’t hesitate to let us know if there are still concerns/questions.
>
> Best regards,
>
> Authors of #1187

---

> ### Author Response · Authors · 2022-12-10
> **Keenly looking forward to your post-rebuttal feedback**
>
> Dear Reviewer Sx5t,
>
> Thanks again for your time in reviewing our work. As the rebuttal discussion is about to end soon, would you mind checking our responses ([a short summary](https://openreview.net/forum?id=9GOjmbRQ2o&noteId=zkh9AXP-H5) and the point-to-point answers) and confirming whether you have any further questions?
>
> If you have any more questions, we are very happy to discuss!
>
> Best regards,
>
> Authors of #1187

---

> ### Author Response · Authors · 2022-12-12
> **Final Call for Your Feedback**
>
> Dear Reviewer Sx5t,
>
> Thanks again for your engagement in reviewing our work. Since we are nearly at the end of the discussion phase, we sincerely hope you can consider positively recommending our work if your concerns are solved. If you still have further comments/suggestions, please don't hesitate to let us know.
>
> Best regards,
>
> Authors of #1187

---

### Author Response · Authors · 2022-11-17
**General Response**

We sincerely thank all reviewers for their efforts and thoughtful feedback. We are encouraged that the majority of the reviewers recognize the novelty and the strong results of our paper.

## Novelty:

Reviewers hvdx, Z771, and y14U recognize the novelty of the paper:
- "The proposed idea is novel and can potentially adapt to different datasets with task-wise numbers of trainable parameters." (Reviewer hvdx)
- "Novelty: the proposed structured and unstructured tuning is new to the parameter-efficient method to the best of my knowledge." (Reviewer hvdx)
- "The idea is novel but the clarity is a concern." (Reviewer Z771)
- "In general, the paper is novel, can be reproduced and is of fair quality." (Reviewer y14U)
- "This paper proposed a novel criterion to efficiently measure the sensitivity (importance) of the pretrained backbone parameters." (Reviewer y14U)

## Strong results:
Reviewers Sx5t, hvdx, Z771 agree that the performance of our method is strong.
- "The results are strong in comparison." (Reviewer Sx5t)
- "The authors show extensive results on different datasets and pretrained models, and the following analyses confirm the effectiveness of the proposed method." (Reviewer hvdx)
- "good results on FGVC and VTAB datasets." (Reviewer Z771)

Before answering the specific questions item by item, we first recap our paper and address two main concerns:


## Recap:
### What is our primary goal?

We aim to develop a visual parameter-efficient tuning approach to tune parameters at **task-specific important positions**.

### Why do we have this goal?

Not all pre-trained parameters contribute equally to the performance for each distinct task.

### How did we achieve this goal?

We propose a **new criterion to efficiently measure the sensitivity (importance) of the pre-trained backbone parameters** to a specific task. Based on the parameter sensitivity, we propose **adaptively determining the tuning granularity for each weight matrix**. Therefore, for the whole model, we structurally update the entire sensitive weight matrices with low-rank reparameterization, and non-structurally tune the weight connections in the insensitive weight matrices, simultaneously.

---

> ### Author Response · Authors · 2022-11-17
> **General Response (Part 2)**
>
> **Major Q1**: What’s our novelty over LoRA [A]?
>
> **A1**: Our method is significantly different from LoRA in twofold.
>
> - LoRA keeps the weight matrices frozen and approximates the update of the entire weight matrix with two low-rank learnable weight matrices. However, similar to the other VPT approaches (e.g., Prompt tuning [B] and Adapter [C]), it neglects the domain gaps across different downstream tasks and adds the low-rank reparameterizations to positions with heuristics (only in the self-attention modules for all downstream tasks). To tackle this fundamental challenge, the main novelty of this paper is identifying the task-specific sensitive positions to tune the weight connections (unstructured tuning) and the low-rank reparameterizations of the entire weight matrices (structured tuning). Therefore, our SPT w/o unstructured tuning is NOT LoRA, as it adds the low-rank reparameterizations to task-specific positions, which includes the weight matrices in the whole model instead of only the self-attention modules. We have already empirically shown that our SPT w/o unstructured tuning achieves better performance than LoRA in Figure 3 (b) (75.4% in the red triangle marker vs. 75.0% in the blue triangle marker) in the original submission.
>
> - LoRA employs only structured tuning that updates entire weight matrices. In contrast, under arbitrary parameter constraints, our SPT integrates both structured and unstructured tuning granularities, and adaptively determines the tuning granularity for each weight matrix according to their proportion of sensitive parameters. In this way, our SPT can achieve higher flexibility and stronger representational capability simultaneously, leading to noticeable gains in parameter-efficiency vs. accuracy trade-off (see Pages 1, 3, 5, 7, and 9 in the initial submission). Accordingly, our SPT achieves better performance while being more scalable than LoRA as shown in Table H. We have added the scalability comparison in Section D of the Appendix in the revised manuscript.
>
> Table H. Comparison of the scalability between our SPT with LoRA [A] on VTAB-1k. LoRA saturates quickly while our SPT has consistent performance gain when the percentage of the trainable parameter increases.
>
> | **Method** | **Tuned / Total** | Natural  | **Specialized** | **Structured** | **Mean** |
> | ---------- | ----------------- | -------- | --------------- | -------------- | -------- |
> | LoRA       | 0.23              | 79.5     | 84.6            | 60.5           | 74.9     |
> | SPT        | 0.30              | **81.7** | **85.5**        | 59.4           | **75.5** |
> | LoRA       | 0.61              | 79.6     | 84.6            | 59.6           | 74.6     |
> | SPT        | **0.53**          | **81.8** | **85.7**        | **60.8**       | **76.1** |
> | LoRA       | 0.69              | 79.8     | 84.9            | 60.2           | 75.0     |
> | SPT        | **0.65**          | **81.8** | **85.9**        | **61.1**       | **76.3** |
>
>
> To make it more clear, we have revised the contribution statements in Section 1 and added discussions with LoRA in Section 3.2 within blue font colour.

---

> > ### Author Response · Authors · 2022-11-17
> > **General Response (Part 3)**
> >
> > **Major Q2**: What is the relationship between our SPT and LoRA [A]?
> >
> > **A2**: Our SPT is orthogonal to the previous VPT methods [A][B][C] that updates the entire weight matrices (like LoRA), as discussed in Major Q1. Therefore, our scheme universally boosts the performance of the VPT methods. In Figure 3 (b) of the initial submission, we have already shown that adding the LoRA [A], Prompt [B], and Adapter [C] modules to the task-specific positions found by our Task-specific Parameter Sensitivity (TPS) can bring solid performance gains. We conduct more experiments to show that further incorporating both structured and unstructured tuning granularities can achieve better performance. The full comparisons are shown in Table I.
> >
> > Table I. Effect of combining our SPT with the other VPT modules on VTAB-1k. Task-specific Parameter Sensitivity (TPS) applies the VPT modules to the sensitive positions found by our criterion. Unstructured tuning also tunes unstructured weight connections for the insensitive weight matrices.
> >
> > | **Method**                        | **Tuned / Total** | Natural  | **Specialized** | **Structured** | **Mean** |
> > | --------------------------------- | ----------------- | -------- | --------------- | -------------- | -------- |
> > | PROMPT-DEEP                       | 1.14              | 78.5     | 82.4            | 55.0           | 72.0     |
> > | PROMPT-DEEP + TPS     | 0.70              | 80.6     | 83.9            | 54.9           | 73.1     |
> > | PROMPT-DEEP + TPS + Unstructured                    | 0.67              | **81.0** | **84.2**        | **55.3**       | **73.7** |
> > | ADAPTER-32                        | 0.71              | 79.6     | 84.0            | 58.3           | 74.0     |
> > | ADAPTER-32 + TPS | 0.54              | 80.8     | **85.5**        | 58.9           | 75.1     |
> > | ADAPTER-32 + TPS + Unstructured                 | 0.55              | **81.4** | 85.3            | **59.3**       | **75.3** |
> > | LoRA-16                           | 0.69              | 79.8     | 84.9            | 60.2           | 75.0     |
> > | LoRA-16 + TPS    | 0.66              | 81.1     | 85.1            | 60.1           | 75.4     |
> > | LoRA-16 + TPS + Unstructured          | 0.61              | **81.6** | **85.6**        | **60.8**       | **76.0** |
> >
> > We have included the corresponding discussions in Section 3.2 in the revised manuscript.
> >
> > **Reference**
> >
> > [A] Edward J Hu, yelong shen, Phillip Wallis, Zeyuan Allen-Zhu, Yuanzhi Li, Shean Wang, Lu Wang, and Weizhu Chen. LoRA: Low-rank adaptation of large language models. In ICLR, 2022.
> >
> > [B] Menglin Jia, Luming Tang, Bor-Chun Chen, Claire Cardie, Serge Belongie, Bharath Hariharan, and Ser-Nam Lim. Visual prompt tuning. In ECCV, 2022.
> >
> > [C] Neil Houlsby, Andrei Giurgiu, Stanislaw Jastrzebski, Bruna Morrone, Quentin De Laroussilhe, Andrea Gesmundo, Mona Attariyan, and Sylvain Gelly. Parameter-efficient transfer learning for nlp. In ICML, pp. 2790–2799, 2019.
> >
> > ## Summary of changes:
> > - We have provided more discussions with LoRA in Section 3.2 and more comparisons with LoRA in terms of scalability in Section D of the Appendix. (Reviewers Sx5t and MqQs)
> > - We have provided more multi-seed results in Table C in the Appendix. (Reviewer hvdx)
> > - We have improved the presentation in Abstract, Sections 1 and 3.2, Figures 1 and 2 to highlight the significance of our paper. (Reviewer MqQs)
> > - We have improved the presentation to describe our method more clearer in Section 3.2. (Reviewers hvdx and y14U)
> > - We have reported the total parameters and inference speed in Table 1. (Reviewer y14U)
> > - We have included the code to reproduce our results in the supplementary material. (Reviewer Z771)
> > - We have provided more cost comparisons and analyses in Table 5. (Reviewer Sx5t)

---

### Author Response · Authors · 2022-12-09
**Keenly looking forward to your post-rebuttal feedback**

Dear Reviewers,

Thanks again for your time in reviewing our work. As the rebuttal discussion is about to end soon, would you mind checking our responses ([a short summary](https://openreview.net/forum?id=9GOjmbRQ2o&noteId=zkh9AXP-H5) and the point-to-point answers) and confirming whether you have any further questions?

If you have any more questions, we are very happy to discuss!


Best regards,

Authors of #1187

---

### Author Response · Authors · 2022-12-10
**Eagerly looking forward to your post-rebuttal feedback**

Dear Reviewers,

Thanks again for your time in reviewing our work. As the rebuttal discussion is about to end soon, would you mind checking our responses ([a short summary](https://openreview.net/forum?id=9GOjmbRQ2o&noteId=zkh9AXP-H5) and the point-to-point answers) and confirming whether you have any further questions?

If you have any more questions, we are very happy to discuss!

Best regards,

Authors of #1187

---

### Author Response · Authors · 2022-12-12
**Dear Reviewers, we believe we have clearly solved your questions. Could you response and raise your ratings?**

Dear Reviewers,

We believe you have received many reminders from us authors. Could you read our response and raise your ratings if we have adequately addressed your questions? **Even one sentence acknowledgment is fine to show some respect to the authors as clearly stated in the reviewer guideline.**

We hope our opinions can be taken into account properly, and we all are committed to developing and maintaining our community well.

Best,

Authors of #1187

---

### Decision · Program_Chairs · 2023-01-20

**Decision:**

Reject

**Justification For Why Not Higher Score:**

All the reviewers unanimously feel the current work needs improvement, and so does the AC.

**Justification For Why Not Lower Score:**

N/A

**Metareview: Summary, Strengths And Weaknesses:**

This submission receives 4 negative reviews and 1 slightly positive review. The major raised issues include memory consumption and computational time, which are not sufficiently addressed by the reviewers. During the post-rebuttal discussion phase, the reviewer who gives the slightly positive review shows that the raised concerns are not well addressed and feels ok to downgrade. Overall, the AC feels all the reviewers are not positive about this submission. The authors shall take these suggestions into account to further improve the current work. Welcome to the next venue.